# UB-SMoE: Universally Balanced Sparse Mixture-of-Experts for Resource-adaptive Federated Fine-tuning of Foundation Models

**Van-Tuan Tran** [1]    **Hong-Hanh Nguyen-Le** [2]    **Marco Ruffini** [1]    **Merim Dzaferagic** [1]

## Abstract

Heterogeneous LoRA-rank methods address system heterogeneity in federated fine-tuning of foundation models by assigning client-specific ranks based on computational capabilities. However, these methods achieve only marginal computational savings, as dense feed-forward computations dominate. Sparse Mixture-of-Experts (SMoE) provides a promising alternative through conditional computation, yet we identify that its naive application to heterogeneous federated settings introduces two critical discordances: (i) expert utilization imbalance and (ii) non-differentiability of Top-K routing. Our convergence analysis demonstrates that these discordances lead to degraded convergence, particularly for resource-constrained clients. To address these challenges, we propose Universally Balanced Sparse Mixture-of-Experts (UB-SMoE), which introduces Dynamic Modulated Routing (DMR) to rebalance expert utilization, and Universal Pseudo-Gradient (PG) to reconstruct learning signals for non-activated experts. These mechanisms form a self-reinforcing cycle that maintains expert viability across heterogeneous clients. Experiments on benchmarks show that UB-SMoE achieves up to $45.0\%$ computational reduction on low-resource clients while improving their performance by $8.7\times$ compared to existing heterogeneous LoRA-rank methods.

## 1. Introduction

Low-Rank Adaptation (LoRA) (Hu et al., 2022) has emerged as a predominant paradigm for federated fine-tuning of large-scale Foundation Models (FMs) (Bian et al., 2025). Specifically, LoRA freezes the pre-trained weights $W_0 \in \mathbb{R}^{d \times l}$ and injects low-rank trainable decomposition matrices $B \in \mathbb{R}^{d \times r}$ and $A \in \mathbb{R}^{r \times l}$. By optimizing only the update $\Delta W = \frac{\alpha}{r} BA$ (where rank $r \ll \min(d, l)$ and $\alpha$ is a scaling factor), LoRA significantly reduces the trainable parameters, making adaptation of large-scale models computationally feasible (Bai et al., 2024; Cho et al., 2024).

While combining LoRA with FL is promising, it is fundamentally challenged by the unique characteristic of distributed networks: **system heterogeneity**. Real-world distributed networks comprise devices with vastly different computational budgets (i.e., low-constrained vs. high-rich devices), implying that *a single model configuration for all devices is ineffective*. We cannot leverage the full computational power of high-end clients if the global model is restricted to a small size enough for edge devices. Conversely, a large, high-capacity model cannot be deployed on low-resource clients due to computational constraints. We need resource-adaptive fine-tuning, where models scale their capacity to match client capabilities.

Existing heterogeneous LoRA-rank methods attempt to address the heterogeneous challenge by assigning client-specific ranks $r_c$ based on computational capabilities (Cho et al., 2024; Bai et al., 2024; Wang et al., 2024b). However, these methods provide only marginal computational savings for resource-constrained clients. The fundamental limitation stems from the architectural mismatch between LoRA adaptation and the dominant computational bottleneck. As shown in Tab. 1, LoRA injects low-rank updates where the adaptation cost scales as $\mathcal{O}(r_c(d + l))$ per layer. However, since $r_c \ll (l \cdot d)$, this cost is negligible compared to the FFN computation of $\mathcal{O}(d \cdot l)$, which remains unchanged regardless of the client's ranks. Consequently, heterogeneous LoRA-rank methods provide only $\sim 5\%$ computation reduction for low-resource clients (Fig. 3 in Appendix) even at the lowest budget level. Furthermore, the merged weights $W_0 + \Delta W$ remain dense during inference, leaving deployment latency unchanged across all client tiers.

Recently, A³SMoE (Tran et al., 2025) integrated Sparse Mixture of Experts (SMoE) (Shazeer et al., 2017) into federated fine-tuning of FMs, offering a *natural mechanism for*

---

[1]School of Computer Science and Statistics, Trinity College Dublin, Ireland [2]School of Computer Science, University College Dublin, Ireland. Correspondence to: Merim Dzaferagic <merim.dzaferagic@tcd.ie>.

*Proceedings of the 43rd International Conference on Machine Learning*, Seoul, South Korea. PMLR 306, 2026. Copyright 2026 by the author(s).

| Aspect | Heterogeneous Sparsity | | Heterogeneous LoRA-rank | | | |
|---|---|---|---|---|---|---|
| | UB-SMoE (ours) | A³SMoE | FLoRA | HetLoRA | FlexLoRA | FLoRIST |
| Client-side computation | $\mathcal{O}(\Gamma \cdot \mathcal{B} \cdot L(M \cdot d + K_c \cdot d \cdot l) + \Gamma \cdot L \cdot M \cdot r(d+l))$ | $\mathcal{O}(\Gamma \cdot \mathcal{B} \cdot L(M \cdot d + K_c \cdot d \cdot l))$ | $\mathcal{O}(\Gamma \cdot \mathcal{B} \cdot L(d \cdot l + r_c(d+l)))$ | $\mathcal{O}(\Gamma \cdot \mathcal{B} \cdot L(d \cdot l + r_c(d+l)) + L \cdot r_c(d+l))$ | $\mathcal{O}(\Gamma \cdot \mathcal{B} \cdot L(d \cdot l + r_c(d+l)))$ | $\mathcal{O}(\Gamma \cdot \mathcal{B} \cdot L(d \cdot l + r_c(d+l)))$ |
| Server-side computation | $\mathcal{O}(C \cdot L \cdot M \cdot r(d+l) + L \cdot M)$ | $\mathcal{O}(C \cdot L \cdot M \cdot r(d+l))$ | $\mathcal{O}(C \cdot L \cdot r_{\max}(d+l))$ | $\mathcal{O}(C \cdot L \cdot r_{\max} \cdot d \cdot l)$ | $\mathcal{O}(C \cdot L(d^2 \cdot l + r_{\max} \cdot d \cdot l))$ | $\mathcal{O}(L \cdot R^2(d \cdot l + R) + L \cdot p \cdot R(d+l))$ |
| Upload per client | $\mathcal{O}(L \cdot M \cdot r(d+l) + L(M+1))$ | $\mathcal{O}(L \cdot M \cdot r(d+l) + L \cdot M)$ | $\mathcal{O}(L \cdot r_c(d+l))$ | $\mathcal{O}(L \cdot r_c(d+l))$ | $\mathcal{O}(L \cdot r_c(d+l))$ | $\mathcal{O}(L \cdot r_c(d+l))$ |
| Download per client | $\mathcal{O}(2 \cdot L \cdot M \cdot r(d+l) + L \cdot M)$ | $\mathcal{O}(L \cdot M \cdot r(d+l))$ | $\mathcal{O}(L \cdot r_c(d+l))$ | $\mathcal{O}(L \cdot r_c(d+l))$ | $\mathcal{O}(L \cdot r_c(d+l))$ | $\mathcal{O}(L \cdot p(d+l))$ |
| Low-resource effective comp. | ✓✓✓ | ✓✓✓ | ✗ | ✗ | ✗ | ✗ |
| Inference adaptability | ✓ | ✓ | ✗ | ✗ | ✗ | ✗ |
| Low-resource performance ($\beta_1$) | **0.3936** | 0.3629 | 0.0094 | 0.0079 | 0.0456 | 0.0112 |
| High-resource performance ($\beta_4$) | **0.5240** | 0.3410 | 0.2996 | 0.4580 | 0.4563 | 0.2724 |
| Average performance | **0.4267** | 0.3861 | 0.1517 | 0.1874 | 0.3303 | 0.1480 |

*Table 1.* Computation and communication complexity analysis between UB-SMoE and other heterogeneous methods. Here $M$ denotes the number of experts, $L$ the number of SMoE layers, $r$ the fixed LoRA rank for SMoE methods, $r_c$ the client-specific rank, $r_{\text{avg}}$ the average rank, $d$ the hidden dimension, $l$ the feed-forward network (FFN) intermediate dimension, $K_c$ the client sparsity, $\Gamma$ local iterations, $\mathcal{B}$ batch size, and $C$ number of clients. For FLoRIST, $R = \sum_c r_c$ and $p$ is the truncated rank. Detailed analysis is provided in Appendix.

*resource adaptability*: high-resource clients activate more experts ($K_{\text{high}}$) while low-resource clients activate fewer ($K_{\text{low}}$) to meet computational budgets. However, naively applying SMoE to heterogeneous federated settings introduces critical optimization discordances:

1. **Expert utilization imbalance**: Experts activated by high-resource clients receive frequent updates and become over-specialized, while those relevant to low-resource clients remain severely under-utilized. This causes a *"rich-get-richer"* dynamic phenomenon.

2. **Non-differentiability of Top-K routing**: Non-activated experts receive zero gradients because the gating function $\gamma_i(x)$ is typically non-zero only for the selected experts. Consequently, low-resource clients with high sparsity leave most experts without learning signals.

Our convergence analysis (Theorem 4.1 in Sec. 4) shows that these discordances introduce a *bias* in the stochastic gradient estimator that creates an **irreducible error floor** in the global objective. This error floor scales inversely with client computational budgets, explaining why resource-constrained clients systematically underperform in federated SMoE systems.

To address these discordances, we propose **Universally Balanced Sparse Mixture-of-Experts (UB-SMoE)**, which reformulates the expert routing paradigm for federated fine-tuning FMs through two novel mechanisms. (i) A **Dynamic Modulated Routing (DMR)** mechanism is introduced to adjust expert selection probabilities based on global utilization statistics. DMR promotes under-utilized experts while preserving top-performing experts for resource-constrained clients, preventing expert collapse. (ii) A **Universal Pseudo-gradient (PG)** mechanism is designed to approximately reconstruct gradients for non-activated experts, ensuring *every expert receives meaningful updates in every round regardless of client sparsity*. These mechanisms create a **self-reinforcing cycle**: PGs maintain expert viability, en-

abling DMR to effectively balance utilization, which in turn generates real gradients that further refine expert parameters.

In summary, our contributions include:

- We identify two critical discordances in federated SMoE fine-tuning and provide a theoretical convergence analysis of their impact on the global convergence. This analysis proves that expert utilization imbalance and gradient sparsity induce an irreducible error floor scaling inversely with client budgets.

- We propose UB-SMoE, a novel method where DMR and PG form a self-reinforcing cycle: PG maintains expert viability through approximate gradients, enabling DMR to effectively balance utilization, which generates real gradients that further refine all experts.

- We validate UB-SMoE on the commonsense reasoning task with 8 datasets (Hu et al., 2023) and the telecommunication domain (Holm, 2021) to show that our method outperforms other federated fine-tuning methods, especially in heterogeneous settings.

## 2. Preliminaries

In this section, we provide formal definitions of Sparse Mixture-of-Experts and heterogeneous federated fine-tuning. For foundational definitions of LoRA and homogeneous federated fine-tuning, we refer readers to Appendix A.

**Definition 1. (Federated Fine-tuning with Heterogeneous LoRA)** Let $\beta_c \in [0, 1]$ denote the computation budget for each client $c$. In a heterogeneous setting, clients utilize client-specific ranks $r_c$, determined by their capability. The rank is defined as $r_c = \lfloor r_{max}\beta_c \rfloor$ ($r_{\min} \leq r_c \leq r_{\max}$). This results in client-specific trainable parameters $\Theta_c = \{B_c, A_c\}$, where $B_c \in \mathbb{R}^{d \times r_c}$ and $A_c \in \mathbb{R}^{r_c \times l}$. The heterogeneous LoRA-rank optimization problem is defined:

$$\min_{\Theta_c} F(\Theta_c) = \sum_{c=1}^{C} p_c F_c(\Theta_c) \qquad (1)$$

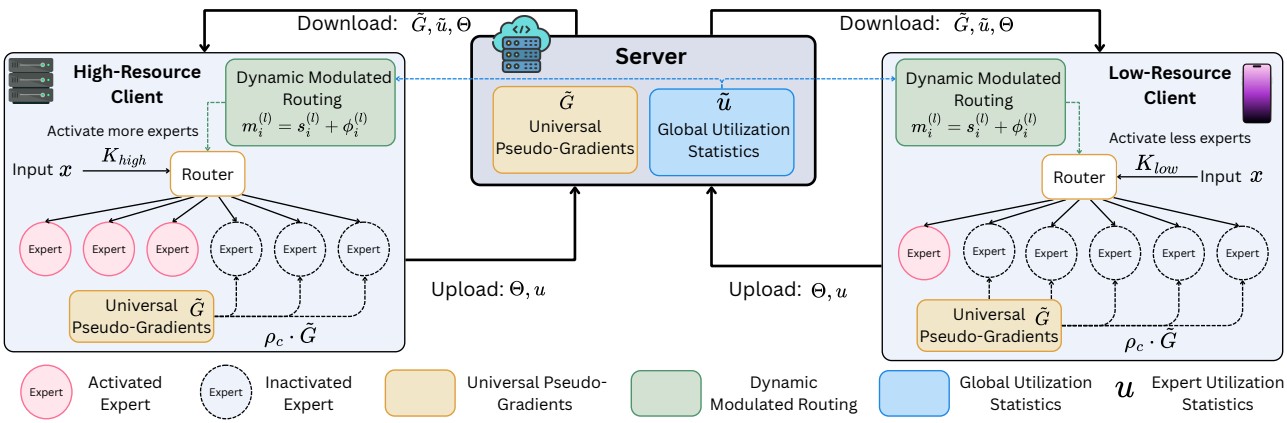

*Figure 1.* Overview of the UB-SMoE architecture for resource-adaptive federated fine-tuning. The system operates in five steps: (1) The server computes global utilization statistics $\tilde{u}$ tracking expert usage across clients; (2) Each client's router applies Dynamic Modulated Routing, modulating logits via $m_i^{(l)} = s_i^{(l)} + \phi_i^{(l)}$ to promote under-utilized but relevant experts; (3) Clients activate a budget-specific number of experts $K_c$; (4) During local training, non-activated experts receive pseudo-gradients scaled by $\rho_c$ to compensate for limited coverage in resource-constrained clients; (5) Clients send parameter updates and utilization statistics back to the server for aggregation.

**Definition 2.** (**Sparse Mixture-of-Experts (SMoE) Layer**). An SMoE layer maps an input $x \in \mathbb{R}^d$ to an output $y \in \mathbb{R}^d$ and is composed of three components: a collection of $M$ expert functions $\{\mathcal{E}_i(\cdot; W_i^{(e)})\}_{i=1}^M$ where each expert $\mathcal{E}_i : \mathbb{R}^d \to \mathbb{R}^d$ is a learnable transformation parameterized by weights $W_i^{(e)}$; a routing function $\mathcal{R}(\cdot; W^{(r)}) : \mathbb{R}^d \to \mathbb{R}^M$ producing affinity scores (logits) $s = \mathcal{R}(x; W^{(r)}) = W^{(r)}x$; and a sparsity parameter $K$ ($1 \leq K \ll M$) specifying the number of experts activated per input.

We define the activation set $\mathcal{A}(x) \subset [M]$ as the indices corresponding to the $K$ largest elements of $s$. The layer output $y$ is computed as: $y = \sum_{i=1}^M \gamma_i(s) \cdot \mathcal{E}_i(x; W_i^{(e)})$, where the gating weight $\gamma_i(s)$ is defined by the softmax function restricted to the activated logits:

$$\gamma_i(s) = \begin{cases} \frac{\exp(s_i)}{\sum_{j \in \mathcal{A}(x)} \exp(s_j)} & \text{if } i \in \mathcal{A}(x) \\ 0 & \text{otherwise.} \end{cases} \quad (2)$$

**Definition 3.** (**Federated Fine-tuning with SMoE-LoRA**) In this paradigm, the SMoE parameters (expert weights $W_i^{(e)}$) and router weights $W^{(r)}$ are adapted using LoRA (Definition 8). Let $\Theta$ denote the collection of all globally shared trainable LoRA parameters for the SMoE layers. Clients have heterogeneous computational capabilities $\{\beta_c\}_{c=1}^C$. Let $K_{max}$ be the maximum allowable sparsity. Each client $c$ activates a client-specific number of experts, $K_c = \lfloor K_{max}\beta_c \rfloor$. In this heterogeneous setting, the federated optimization problem is formulated as:

$$\min_{\Theta} F(\Theta) = \sum_{c=1}^C p_c F_c(\Theta; K_c) \quad (3)$$

where $F_c(\Theta; K_c)$ denotes the local objective function evaluated using the shared parameters $\Theta$ under the client-specific sparsity constraint $K_c$.

## 3. SMoE in Heterogeneous Federated Fine-Tuning

While SMoE architectures have demonstrated remarkable success in centralized settings (Sun et al., 2024; Zoph et al., 2022), their direct application to heterogeneous federated fine-tuning presents fundamental challenges. In this section, we identify two critical discordances that contribute to performance degradation.

### 3.1. Discordance 1: Expert Utilization Imbalance

The expert utilization imbalance in federated fine-tuning manifests at two distinct levels: local imbalance within individual clients and global imbalance across the federated system.

**Definition 4.** (**Per-Round Expert Utilization Rates**) For a federated system with $C$ clients and an SMoE layer with experts $\mathcal{E} = \{1, \dots, M\}$:

- **Local utilization rate**: $U_{c,i}^{local} = \frac{\sum_{x \in \mathcal{D}_c} \mathbb{I}(i \in \mathcal{A}_c(x))}{|\mathcal{D}_c| \cdot K_c}$,

- **Global utilization rate**: $U_i^{global} = \sum_{c=1}^C p_c \cdot U_{c,i}^{local}$,

where $\mathcal{A}_c(x)$ denotes the $K_c$ activated experts for input $x$ at client $c$, $K_c = \lfloor K_{max}\beta_c \rfloor$ is the number of experts client $c$ can activate based on its budget $\beta_c$.

Different from the centralized setting (Zoph et al., 2022; Qiu et al., 2025), expert utilization imbalance in federated

fine-tuning manifests at two levels. **Local imbalance** occurs when routers preferentially select certain experts, leading to parameter redundancy. **Global imbalance** emerges from heterogeneous computational budgets: high-resource clients (larger $\beta_c$, thus higher $K_c$) develop well-trained experts, while low-resource clients maintain under-trained experts with limited coverage. These effects compound, making the utilization imbalance significantly more severe in federated settings than in centralized training.

### 3.2. Discordance 2: Non-Differentiability of Top-K Routing

The Top-K routing mechanism introduces a fundamental optimization challenge through its discrete selection process (Liu et al., 2023; Panda et al., 2024). In heterogeneous settings where $|\mathcal{A}_c(x)| = K_c$, the gradient with respect to router parameters $W^{(r)}$ is:

$$\nabla_{W^{(r)}}\mathcal{L}_c = \sum_{i \in \mathcal{A}_c(x)} \frac{\partial \mathcal{L}_c}{\partial \gamma_i} \nabla_{W^{(r)}} \gamma_i. \qquad (4)$$

Since non-activated experts have $\gamma_i = 0$ (Definition 2), they receive zero gradients. Only $K_c$ out of $M$ experts provide learning signals, creating an **exploration-exploitation imbalance**: without gradient signals from non-activated experts, the router exploits currently preferred experts rather than exploring optimal alternatives. It is even exacerbated with low-resource clients, where high expert coverage is unavailable due to computational constraints.

The non-differentiability of the Top-K routing function becomes particularly severe in federated environments where clients activate different numbers of experts based on their computational capabilities. For a client $c$ with computational budget $\beta_c$ activating $K_c = \lfloor K_{\max}\beta_c \rfloor$ experts per input, the expected number of gradient updates per expert is proportional to $\frac{K_c}{M}\Gamma_c$, where $\Gamma_c$ denotes the local training iterations. Since low-resource clients activate fewer experts ($K_c^{\text{low}} < K_c^{\text{high}}$), their experts receive disproportionately fewer gradient updates, resulting in slower convergence compared to high-resource clients.

## 4. Convergence Analysis

We analyze client-specific convergence in heterogeneous SMoE settings, demonstrating how the two discordances: expert utilization imbalance and non-differentiability of Top-K routing, lead to degraded convergence for resource-constrained clients.

Our analysis builds upon the established field of optimization with biased gradient estimators. While standard Stochastic Gradient Descent (SGD) typically relies on unbiased gradient estimates (Bottou et al., 2018), there are scenarios that often introduce bias, such as compression,

quantization (Ajalloeian & Stich, 2020; Hu et al., 2021; Karimi et al., 2019), or, as in this case, **conditional computation** in SMoE. In the context of federated SMoE, the optimization process faces two primary challenges that affect the gradient estimation: high variance due to sparse expert activation and bias introduced by the discrete routing mechanism. We first introduce the crucial definitions to characterize the gradient estimation process under sparse activation.

### 4.1. Gradient Bias in Sparse MoE

Let $\Theta$ denote the collection of all trainable LoRA parameters. We decompose $\Theta$ into parameters specific to the experts and the router: $\Theta = {\Theta_i^{(e)}}_{i=1}^M \cup \Theta^{(r)}$, where $\Theta_i^{(e)}$ are the LoRA parameters adapting the expert weights $W_i^{(e)}$, and $\Theta^{(r)}$ adapts the router weights $W^{(r)}$. The optimization objective is $F(\Theta)$ (Eq. 3).

**Definition 5. (Sparse Stochastic Gradient).** Consider an SMoE layer at client $c$. For an input $x$, the stochastic gradient with respect to the parameters $\Theta_i^{(e)}$ of expert $i$ under the Top-$K_c$ routing mechanism is:

$$g_{c,i}(\Theta; x) = \mathbb{I}(i \in \mathcal{A}_c(x)) \cdot \nabla_{\Theta_i^{(e)}}\mathcal{L}_c(\Theta; x), \qquad (5)$$

where $\mathcal{L}_c$ is the local loss, and $\mathcal{A}_c(x)$ is the set of $K_c$ activated experts. Non-activated experts receive zero gradients.

**Definition 6. (Expert Activation Probability and Conditional Gradient).** Let $p_{c,i}(\Theta) = P_{x \sim \mathcal{D}_c}(i \in \mathcal{A}_c(x))$ be the probability that expert $i$ is activated by client $c$. We define the conditional expected gradient for $\Theta_i^{(e)}$ as:

$$\nabla F_{c|i}(\Theta) = \mathbb{E}_{x \sim \mathcal{D}_c}[\nabla_{\Theta_i^{(e)}}\mathcal{L}_c(\Theta; x)|i \in \mathcal{A}_c(x)], \qquad (6)$$

where $F_{c|i}(\Theta)$ is the average local loss calculated only over the data $x$ from client $c$ that activates expert $i$. The expected sparse gradient estimator is $\mathbb{E}[g_{c,i}(\Theta)] = p_{c,i}(\Theta) \cdot \nabla F_{c|i}(\Theta)$.

The difference between this expected sparse gradient and the true local gradient characterizes the optimization bias introduced by the routing mechanism.

**Definition 7. (Gradient Estimation Bias).** The bias of the sparse gradient estimator with respect to $\Theta_i^{(e)}$ at client $c$ is: $B_{c,i}(\Theta) = \nabla_{\Theta_i^{(e)}}F_c(\Theta) - \mathbb{E}[g_{c,i}(\Theta)]$. The aggregate bias across the federated system is $B^e(\Theta) = \sum_{c=1}^C p_c B_c^{(e)}(\Theta)$.

The bias arises because the sparse gradient only reflects the loss landscape conditional on the expert being active. If an expert is rarely activated (low $p_{c,i}(\Theta)$), the estimator deviates significantly from the true gradient.

## 4.2. Convergence Analysis

We rely on standard assumptions used in the convergence analysis of FL (Karimireddy et al., 2020; Li et al., 2019a; Li & Lyu, 2023; Koloskova et al., 2020) and biased optimization (Hu et al., 2021; Demidovich et al., 2023; Ajalloeian & Stich, 2020), adapted to the LoRA parameters space $\Theta$.

**Assumption 1.** ($L$-**Smoothness**). The local objective functions $F_c(\Theta)$ are $L$-smooth for all $\Theta_1, \Theta_2$, $||\nabla F_c(\Theta_1) - \nabla F_c(\Theta_2)|| \leq L||\Theta_1 - \Theta_2||$.

This assumption is standard in FL convergence analysis (Li et al., 2019b; Li & Lyu, 2023).

**Assumption 2.** (**PL Condition**) The global objective function $F(\Theta)$ satisfies the Polyak-Łojasiewicz condition with constant $\mu > 0$. That is, $\frac{1}{2}||\nabla F(\Theta)||^2 \geq \mu(F(\Theta) - F(\Theta^*))$, where $\Theta^*$ is the global optimum.

The PL condition is a standard local regularity assumption in non-convex FL analyses, enabling convergence-rate characterization without strong convexity (Ying et al., 2026; Liu & Zhou, 2025; Bao et al., 2025). In our setting, the full foundation model is frozen, and optimization is restricted to lightweight LoRA adapters, so the trainable search space is substantially lower-dimensional than full-model training. Therefore, we assume PL regularity only for the adapter optimization dynamics, not for the full transformer loss.

**Assumption 3.** (**Bounded Variance and Gradients (Bottou et al., 2018; Gower et al., 2019)**). The variance of the stochastic sparse gradients (Definition 5) is bounded by $\sigma^2$. The expected squared norm of the stochastic gradients is bounded: $\mathbb{E}[||g_c(\Theta)||^2] \leq G^2$.

The bounded-gradient assumption is widely used in convergence analyses of heterogeneous and model-adaptive FL (Zhou et al., 2023; Han et al., 2024; Lan et al., 2024; Zong et al., 2025). We use it to isolate the additional bias introduced by sparse Top-$K$ routing, which is the phenomenon studied in Theorem 4.1.

**Assumption 4.** (**Bounded Gradient Divergence**). We assume bounded divergence between the conditional expected gradient and the true local gradient: $||\nabla F_{c|i}(\Theta) - \nabla_{\Theta_i^{(e)}} F_c(\Theta)||^2 \leq \delta^2$.

The assumption 4 characterizes the degree of utilization of the experts and the distribution of the data routed to them. It is similar to bounding data heterogeneity in standard FL analysis (Li et al., 2019a), constraining how much the data distribution activating a specific expert differs from the overall local distribution $\mathcal{D}_c$. We can quantify its impact on the bias. Using the triangle inequality: $||B_{c,i}(\Theta)|| \leq (1 - p_{c,i}(\Theta))||\nabla_{\Theta_i^{(e)}} F_c(\Theta)|| + p_{c,i}(\Theta)\delta$. If $\delta^2$ is small, the magnitude of the bias is primarily driven by the activation probability: $||B_{c,i}(\Theta)|| \approx (1 - p_{c,i}(\Theta))||\nabla_{\Theta_i^e} F_c(\Theta)||$.

**Assumption 5.** (**Lipschitz Continuity of Bias**). The aggregate bias function $B(\Theta) = \nabla F(\Theta) - \sum_c p_c\mathbb{E}[g_c(\Theta)]$ is $L_B$-Lipschitz. That is, $||B(\Theta_1) - B(\Theta_2)|| \leq L_B||\Theta_1 - \Theta_2||$. Let $L_{eff} = L + L_B$.

Assumption 5 is crucial for the convergence analysis of SGD with biased gradients (Hu et al., 2021; Ajalloeian & Stich, 2020). In the context of SMoE, although the routing function (Top-K) is discontinuous, the bias depends on the *expected sparse gradient*, which is determined by the activation probabilities $p_{c,i}(\Theta)$. This assumption implies that the expectation over the data distribution smooths out the discontinuities of the Top-K operation. Furthermore, we require the curvature to dominate the bias sensitivity, specifically $\mu^2 > 4L_B^2$ (Ajalloeian & Stich, 2020).

We now present Theorem 4.1, demonstrating that the use of biased sparse gradients leads to an error floor.

**Theorem 4.1.** (*Global Convergence Rate under Heterogeneous Sparsity*). *Under Assumptions 1-5, consider the federated SMoE fine-tuning process where clients perform $\Gamma$ local updates using the biased sparse gradient estimator with a decaying learning rate $\eta_t = \frac{1}{\sqrt{\Gamma(t+1)}}$. Let $\Theta^*$ be the global optimum of $F(\Theta)$. The expected optimality gap after $T$ communication rounds satisfies:*

$$\mathbb{E}[F(\Theta^T)] - F(\Theta^*) \leq \underbrace{\mathcal{O}\left(\frac{G^2(L + L_{eff}^2)}{\mu'\sqrt{T\Gamma}}\right)}_{\text{Optimization Error}} \quad (7)$$
$$+ \underbrace{B_{SMoE}}_{\text{Bias Error}},$$

*where the bias error term is given by $B_{SMoE} = \frac{2||B(\Theta^*)||^2}{\mu'}$, and $\mu' = \mu - \frac{4L_B^2}{\mu} > 0$ is the bias-penalized curvature.*

The proof can be found in Appendix B. Theorem 4.1 formally shows that naive application of SMoE in federated settings leads to convergence to only to a neighborhood of the optimum. In this analysis, we decompose the error into two components: (i) Optimization Error, which depends on the gradient bound $G^2$ and client drift (captured by $L_{eff}^2$), decaying at a rate of $\mathcal{O}(1/\sqrt{T})$; and (ii) Bias Error $B_{SMoE}$, which depends on the magnitude of the aggregate bias at the optimum. This bias stems directly from the gradient sparsity introduced by the Top-K mechanism (Discordance 2).

**Corollary 1.** (*Impact of Client Sparsity and Utilization Imbalance on Convergence Error*). *The contribution of client $c$ to the aggregate bias error floor $B_{SMoE}$ is inversely related to its computational budget $K_c$ and is exacerbated by expert utilization imbalance. Under Assumption 4 ($\delta \approx 0$), the squared norm of the local bias $||B_c^{(e)}(\Theta^*)||^2$ is approxi-*

*mately:*

$$\|B_c^{(e)}(\Theta^*)\|^2 \approx \sum_{i=1}^{M}(1-p_{c,i}(\Theta^*))^2\|\nabla_{\Theta_i^{(e)}}F_c(\Theta^*)\|^2. \quad (8)$$

*Let $G_{c,\min}^2$ and $G_{c,\max}^2$ be the lower and upper bounds, respectively, on the individual expert gradient norms at the optimum, such that $G_{c,\min}^2 \leq \|\nabla_{\Theta_i^{(e)}}F_c(\Theta^*)\|^2 \leq G_{c,\max}^2$ for all i. Under the constraint $\sum_i p_{c,i} = K_c$ and $p_{c,i} \in [0,1]$, the bias is bounded as follows:*

$$G_{c,min}^2 \cdot M\left(1 - \frac{K_c}{M}\right)^2 \leq \|B_c^{(e)}(\Theta^*)\|^2 \leq G_{c,max}^2 \cdot (M - K_c). \quad (9)$$

The lower bound is achieved under perfect utilization balance ($p_{c,i} = K_c/M$), minimizing the convex sum in Eq. 8. The upper bound is achieved under maximal imbalance (where $M - K_c$ experts are never utilized, $p_{c,i} = 0$), maximizing the convex sum. When $K_c$ is small (i.e., resource-constrained clients), the potential bias is large, demonstrating that resource-constrained clients inherently introduce significant bias into the global optimization process, especially when utilization is imbalanced (Discordance 1).

# 5. Methodology: UB-SMoE

This section details **Universally Balanced Sparse Mixture-of-Experts (UB-SMoE)**, which is designed to resolve the optimization discordances in heterogeneous federated fine-tuning (Fig. 1). UB-SMoE introduces two mechanisms: (i) **Dynamic Modulated Routing (DMR)** addresses expert utilization imbalance; and (ii) **Universal Pseudo-Gradient (PG)** reconstructs learning signals for non-activated experts.

## 5.1. Dynamic Modulated Routing (DMR)

DMR regulates expert selection using global utilization statistics through learnable modulation parameters.

**Modulated Logit Computation.** For an SMoE layer $l \in \{1, \ldots, L\}$ with $M$ experts, we introduce a learnable modulation vector $\phi^{(l)} = [\phi_1^{(l)}, \ldots, \phi_M^{(l)}] \in \mathbb{R}^M$, initialized to zero. To prevent utilization biases from overriding semantic relevance, our router integrates two distinct control signals: the **input signal** (data condition) and the **utilization signal** (load balancing). Instead of directly applying Top-K selection on raw affinity scores $\mathbf{s}^{(l)} = W^{(r)}x$, we first identify a candidate set $\mathcal{T}^{(l)}$ comprising the top $N_p$ experts with highest raw scores ($K_{max} \leq N_p \ll M$). We set $N_p = 2$ (see ablation in Tab. 21 in Appendix). Then, we apply the learnable modulation parameters $\phi^{(l)}$ only to the experts in $\mathcal{T}^{(l)}$. The modulated logits $m^{(l)}$ are computed as:

$$m_i^{(l)} = \begin{cases} s_i^{(l)} + \phi_i^{(l)} & \text{if } i \in \mathcal{T}^{(l)} \\ s_i^{(l)} & \text{otherwise} \end{cases}, \quad (10)$$

The client selects active experts $\mathcal{A}_c(\mathbf{x}) = \text{Top-K}(p^{(l)}, K_c)$ where $p^{(l)} = \text{softmax}(m^{(l)})$.

**Regularization.** To prevent the modulation parameters from diverging, we apply $L_2$ regularization during client training to constrain $\phi^{(l)}$ within the range $[\phi_{min}, \phi_{max}]$:

$$\mathcal{L}_{reg} = \lambda\left(\|\text{ReLU}(\phi_{min} - \phi^{(l)})\|_2^2 + \|\text{ReLU}(\phi^{(l)} - \phi_{max})\|_2^2\right) \quad (11)$$

**Utilization-aware Global Update.** The server computes the global utilization rate $\tilde{u}_i^{(l)}$ for each expert $i$:

$$\tilde{u}_i^{(l)} = \sum_{c=1}^{C} p_c \frac{a_{c,i}^{(l)}}{n_c^{(l)}}, \quad (12)$$

where $a_{c,i}^{(l)}$ is the activation count, $n_c^{(l)}$ is the total tokens processed, and $p_c = |D_c|/\sum_{c'}|D_{c'}|$ is the client weight. We define a target uniform utilization $u^* = \bar{K}/M$, where $\bar{K} = \sum p_c K_c$ is the average system sparsity. The update rule penalizes over-utilized experts and boosts under-utilized ones:

$$\tilde{\phi}_i^{(l)} = \tanh\left(\frac{u^*}{\tilde{u}_i^{(l)} + \epsilon} - 1\right) \quad (13)$$

$$\phi_i^{(l),(t+1)} = (1 - \zeta)\tilde{\phi}_i^{(l)} + \zeta\phi_i^{(l),(t)} \quad (14)$$

where $\zeta \in [0,1]$ is a momentum factor and the $\tanh$ function ensures bounded modulation values.

## 5.2. Universal Pseudo-Gradient (PG)

DMR balances selection, but it does not solve the non-differentiability of Top-K. In heterogeneous FL, low-resource clients ($K_c \ll M$) produce zero gradients for most experts, creating optimization blind spots. We propose a universal pseudo-gradient (PG) mechanism to reconstruct meaningful update directions for non-activated experts.

**Pseudo-gradient Computation.** After aggregating updates at round $t + 1$, the server computes a global pseudo-gradient vector $\tilde{G}^{(l)}$ that captures the parameter evolution:

$$\tilde{G}_i^{(l),(t+1)} = \frac{\Theta_i^{(e),(l),(t)} - \Theta_i^{(e),(l),(t+1)}}{\eta \cdot \Gamma}, \quad (15)$$

where $\eta > 0$ is the local learning rate and $\Gamma$ is the number of local iterations.

**Resource-aware Gradient Injection.** During local training, clients use this global signal to update experts they did not select. The modified gradient estimator $\hat{g}_{c,i}$ for client $c$ is:

$$\hat{g}_{c,i}(\Theta; \mathbf{x}) = \begin{cases} \nabla_{\Theta_i^{(e)}}\mathcal{L}_c(\Theta; \mathbf{x}) & \text{if } i \in \mathcal{A}_c(\mathbf{x}) \\ \rho_c \cdot \tilde{G}_i^{(l),(t)} & \text{if } i \notin \mathcal{A}_c(\mathbf{x}) \end{cases}, \quad (16)$$

*Table 2.* **Performance comparison on Commonsense-15K benchmark using OLMoE-1B-7B**. Performance averaged over all computation budgets. Higher values indicate better performance. Bold values highlight the best performance.

| Dataset | Heterogeneous LoRA-rank methods | | | | Heterogeneous sparsity methods | | |
|---|---|---|---|---|---|---|---|
| | HetLoRA | FlexLoRA | FLoRA | FLoRIST | SMoE-LLB | A$^3$SMoE | UB-SMoE |
| ARC-Challenge | 0.1284 | 0.3012 | 0.0868 | 0.0960 | 0.3080 | 0.3347 | **0.3611** |
| ARC-Easy | 0.1582 | 0.4136 | 0.1004 | 0.1097 | 0.4213 | 0.4724 | **0.5017** |
| BoolQ | 0.3709 | 0.4573 | 0.3472 | 0.3407 | **0.5122** | 0.4301 | 0.4952 |
| HellaSwag | 0.0714 | 0.1607 | 0.0674 | 0.0605 | 0.2096 | **0.2448** | 0.2258 |
| OpenBookQA | 0.1200 | 0.3225 | 0.1160 | 0.1380 | 0.3360 | 0.3925 | **0.4030** |
| PIQA | 0.2278 | 0.3327 | 0.2285 | 0.2319 | 0.4244 | 0.4219 | **0.5118** |
| Social IQa | 0.1452 | 0.3099 | 0.0885 | 0.0814 | 0.3801 | 0.4193 | **0.4486** |
| WinoGrande | 0.2770 | 0.3445 | 0.1786 | 0.1257 | 0.4410 | 0.3733 | **0.4665** |
| Average | 0.1874 | 0.3303 | 0.1517 | 0.1480 | 0.3791 | 0.3861 | **0.4267** |

where $\rho_c = \sqrt{\bar{K}/K_c}$ denotes the resource-adaptive scaling factor. This factor amplifies the pseudo-gradient signal for resource-constrained clients (small $K_c$). As shown in our theoretical analysis, the gradient bias is inversely proportional to $K_c$. By scaling up the pseudo-gradient, we compensate for the limited expert coverage of low-resource clients, ensuring their local models remain aligned with the global consensus.

## 6. Experiments

**Benchmarks.** We evaluate our method on Commonsense-15K benchmark (Hu et al., 2023) (contains 8 distinct commonsense reasoning datasets), TeleQuAD (in telecommunication domain) (Holm, 2021).

**Baselines.** We compare UB-SMoE with two groups of methods for handling system heterogeneity: (1) heterogeneous LoRA-rank methods, including HetLoRA (Cho et al., 2024), FlexLoRA (Bai et al., 2024), FLoRA (Wang et al., 2024b), FLoRIST (Ramesh & Dass, 2025); and (2) Heterogeneous sparsity methods, including A$^3$SMoE (Tran et al., 2025) and the same MoE model enabled with local load-balancing (LLB-SMoE) using auxiliary loss (Zoph et al., 2022).

**Implementation.** We use OLMoE-1B-7B (Muennighoff et al., 2024) and OLMo-1B (Groeneveld et al., 2024) as the base models for all experiments. The maximum number of activated experts is $K_{max} = 8$. For heterogeneous sparsity methods, we fix LoRA rank at $r = 20$ across all clients with varying activated experts $K_c \in \{1, 2, 4, 8\}$. Regarding heterogeneous LoRA-rank methods, client-specific ranks are set to $r_c \in \{6, 8, 12, 20\}$. Detailed experimental configurations are provided in Appendix E.

### 6.1. Performance Across Benchmarks

Tables 2-3 report results on the Commonsense-15K benchmark, averaged over all computation budgets: Table 2 com-

*Table 3.* Performance comparison on Commonsense-15K benchmark using dense OLMo-1B. Heterogeneous LoRA-rank methods use client-specific ranks $r_c \in \{12, 16, 24, 40\}$ during training. UB-SMoE is evaluated at budget $\beta_4$ with matched trainable activated parameters. Higher values indicate better performance. Bold values highlight the best performance within each budget category.

| Dataset | FlexLoRA | HetLoRA | FLoRA | FLoRIST | UB-SMoE |
|---|---|---|---|---|---|
| ARC-Challenge | 0.2654 | 0.2159 | 0.1212 | 0.0734 | **0.5333** |
| ARC-Easy | 0.2740 | 0.2180 | 0.1077 | 0.0614 | **0.7184** |
| BoolQ | 0.5101 | **0.5523** | 0.2651 | 0.1917 | 0.4697 |
| HellaSwag | 0.1358 | 0.1104 | 0.1911 | 0.1294 | **0.3536** |
| OpenBookQA | 0.2660 | 0.2340 | 0.1000 | 0.0780 | **0.6320** |
| PIQA | 0.4777 | 0.5005 | 0.4918 | 0.4913 | **0.6931** |
| Social IQa | 0.3557 | 0.3362 | 0.1029 | 0.0583 | **0.6039** |
| WinoGrande | 0.4775 | 0.4854 | 0.2928 | 0.2092 | **0.5043** |
| Average | 0.3453 | 0.3316 | 0.2091 | 0.1616 | **0.5636** |

pares methods on the SMoE backbone while Table 3 further compares methods on OLMo-1B backbone. For heterogeneous LoRA-rank baselines, we configure client-specific ranks as $r_c \in \{12, 16, 24, 40\}$ corresponding to budget levels $\beta_1$ through $\beta_4$ during training. Note that these methods produce dense models at inference time, as the heterogeneous LoRA adapters are aggregated into full-rank updates. To ensure fair comparison, we evaluate UB-SMoE at a budget $\beta_4$, such that the number of trainable activated parameters similarly matches across methods.

On OLMoE-1B-7B backbone, UB-SMoE achieves the highest average performance of 0.4267, outperforming the strongest heterogeneous sparsity baseline, A$^3$SMoE, by 10.5%, and the strongest heterogeneous LoRA-rank baseline, FlexLoRA, by 29.1%. On OLMo-1B. backbone, UB-SMoE achieves an average accuracy of 0.5636, substantially outperforming all heterogeneous LoRA-rank baselines. For example, UB-SMoE surpasses FlexLoRA by 63.2%, HetLoRA by 70.0%, FLoRA by 169.5%, and FLoRIST by 248.6%. These results indicate that UB-SMoE is not only effective under sparse expert fine-tuning, but also provides a stronger accuracy-resource trade-off than dense-model

*Table 4.* **Performance comparison across different resource budgets on Commonsense-15K**. Higher values indicate better performance. Bold values highlight the best performance within each budget category.

| Budget | Heterogeneous Setting | | | | | | | Homogeneous Setting | | | | |
| | LoRA-rank Methods | | | | Sparsity Methods | | | LoRA-rank Methods | | | | |
| | HetLoRA | FlexLoRA | FLoRA | FLoRIST | SMoE-LLB | A³SMoE | UB-SMoE | FedEx-LoRA | FFA-LoRA | FedIT | FedSB | UB-SMoE |
|---|---|---|---|---|---|---|---|---|---|---|---|---|
| $\beta_1$ | 0.0079 | 0.0456 | 0.0094 | 0.0112 | 0.3531 | 0.3629 | **0.3936** | 0.3367 | 0.2432 | **0.3740** | 0.1354 | 0.3055 |
| $\beta_2$ | 0.0699 | 0.2818 | 0.0480 | 0.0538 | 0.3847 | **0.4310** | 0.3359 | 0.3692 | 0.3512 | **0.4037** | 0.1622 | 0.3587 |
| $\beta_3$ | 0.2137 | 0.5375 | 0.2497 | 0.2545 | 0.3961 | 0.4096 | **0.4716** | 0.3741 | 0.3733 | 0.4382 | 0.2576 | **0.5004** |
| $\beta_4$ | 0.4580 | 0.4563 | 0.2996 | 0.2724 | 0.3824 | 0.3410 | **0.5240** | 0.4337 | 0.3851 | 0.4503 | 0.3318 | **0.5636** |
| Average | 0.1874 | 0.3303 | 0.1517 | 0.1480 | 0.3791 | 0.3861 | **0.4313** | 0.3784 | 0.3382 | 0.4165 | 0.2217 | **0.4320** |

*Table 5.* **Performance comparison across different resource budgets on TeleQuAD telecommunications question-answering benchmark**. Models are evaluated under both IID and non-IID client data distributions using BERTScore F1. Higher values indicate better performance, and bold values highlight the best result within each budget and data-distribution setting.

| Budget | IID | | | | | Non-IID | | | | |
| | HetLoRA | FlexLoRA | SMoE-LLB | A³SMoE | UB-SMoE | HetLoRA | FlexLoRA | SMoE-LLB | A³SMoE | UB-SMoE |
|---|---|---|---|---|---|---|---|---|---|---|
| $\beta_1$ | 39.35 | 39.02 | 48.31 | **50.76** | 46.25 | 35.84 | 37.73 | 44.56 | 37.24 | **47.66** |
| $\beta_2$ | 44.87 | 47.58 | **61.15** | 61.04 | 58.82 | 40.76 | 44.13 | 59.68 | 41.71 | **60.04** |
| $\beta_3$ | 54.93 | 58.68 | 65.09 | 65.06 | **65.83** | 44.63 | 42.87 | 65.98 | 41.33 | **66.31** |
| $\beta_4$ | 58.05 | 62.65 | 65.41 | 65.63 | **67.60** | 46.26 | 42.10 | 68.23 | 42.52 | **68.29** |
| Average | 49.30 | 51.98 | 59.99 | **60.62** | 59.63 | 41.87 | 41.71 | 59.61 | 40.70 | **60.58** |

LoRA-rank adaptation under matched trainable-parameter budgets. We attribute this improvement to two complementary effects: dynamic modulated routing improves expert selection under heterogeneous sparsity, while universal pseudo-gradients provide learning signals to experts that are not locally activated by resource-constrained clients.

### 6.2. Performance Across Resource Budgets

Tables 4 and 5 evaluate performance across four computation budgets, $\beta_1$–$\beta_4$, on Commonsense-15K and TeleQuAD, respectively. The Commonsense-15K results report average accuracy over eight reasoning tasks, while TeleQuAD reports BERTScore F1 under IID and non-IID client data distributions.

On Commonsense-15K, Table 4 shows that heterogeneous LoRA-rank methods are highly sensitive to tight resource budgets. At $\beta_1$, HetLoRA, FLoRA, and FLoRIST achieve only 0.0079, 0.0094, and 0.0112 average accuracy, respectively. In contrast, sparsity-based methods remain effective, with SMoE-LLB, A³SMoE, and UB-SMoE achieving 0.3531, 0.3629, and 0.3936. UB-SMoE obtains the best performance among heterogeneous methods at both $\beta_1$ and $\beta_4$, demonstrating strong adaptation across resource budgets.

Table 5 further evaluates resource adaptation on TeleQuAD, a domain-specific telecommunications question-answering benchmark. Under IID data, UB-SMoE is competitive at low budgets and achieves the best performance at larger budgets, reaching 65.83 and 67.60 BERTScore F1 at $\beta_3$ and $\beta_4$. Under non-IID data, UB-SMoE achieves the best result

at every budget, improving from 47.66 at $\beta_1$ to 68.29 at $\beta_4$.

Overall, these results show that UB-SMoE provides a strong accuracy–resource trade-off across benchmarks, computation budgets, and data distributions. Its advantage is most pronounced in constrained or non-IID settings, where balanced expert utilization is critical for stable federated adaptation.

### 6.3. Analysis

#### 6.3.1. CONTRIBUTION OF UB-SMoE MECHANISMS

Tab. 6 quantifies the contributions of our two primary mechanisms: DMR (Sec. 5.1) and PG (Sec. 5.2). The PG mechanism increases the FL system's performance from 0.1701 to 0.2659 by providing learning signals to both activated and non-activated experts. With additional $\phi$ regularization and utilization-aware update, the utilization imbalance problem has been addressed, and the performance increases to 0.4267.

*Table 6.* Ablation study quantifying the contribution of each core mechanism of UB-SMoE. The results are averages across 8 commonsense reasoning datasets.

| Pseudo-gradients | Dynamic Modulated Routing | | Avg. |
| | $\phi$ Regularization | Utilization-aware Update | |
|---|---|---|---|
| | | | 0.1701 |
| ✓ | | | 0.2659 |
| ✓ | ✓ | | 0.2839 |
| ✓ | | ✓ | 0.3591 |
| | ✓ | ✓ | 0.4009 |
| ✓ | ✓ | ✓ | **0.4267** |

### 6.3.2. SCALABILITY ANALYSIS

To evaluate scalability with larger client populations and diverse heterogeneity patterns, we conduct experiments with 32 clients under three configurations: (1) uniform distribution with 50% participation rate, (2) long-tail distribution where 75% of clients are low-resource, and (3) reverse long-tail where 75% are high-resource. Table 7 shows that UB-SMoE maintains superior performance across all heterogeneity patterns, demonstrating robustness to both client scale and resource distribution. Full per-dataset results are in Appendix G.2.

*Table 7.* Scalability to 32 clients under different heterogeneity patterns on Commonsense-15K benchmark. Performance averaged over all computation budgets.

| Pattern | HetLoRA | FlexLoRA | A$^3$SMoE | SMoE-LLB | Ours |
|---|---|---|---|---|---|
| Uniform (50%) | 0.1505 | 0.2070 | 0.3998 | 0.3928 | **0.4036** |
| Long-tail | 0.1832 | 0.2585 | 0.2988 | 0.2961 | **0.3047** |
| Rev. Long-tail | 0.2242 | 0.2601 | 0.4148 | 0.3928 | **0.4272** |

### 6.3.3. EXPERT UTILIZATION BALANCE

To quantify how effectively PG addresses expert utilization imbalance, we compute the global utilization entropy: $H_{\text{global}} = -\sum_{i=1}^{M} \tilde{u}_i \log(\tilde{u}_i)$. We also provide an analysis on Gini coefficients in Sec. G.6 of Appendix.

From Fig. 2, we observe that: (i) UB-SMoE achieves the highest mean entropy ($H \approx 6.5$) with lower variance; (ii) UB-SMoE progressively improves utilization balance via communication rounds. Per-layer analysis in Appendix G.7 confirms that UB-SMoE maintains superior entropy across all 16 SMoE layers, while baselines exhibit severe imbalance.

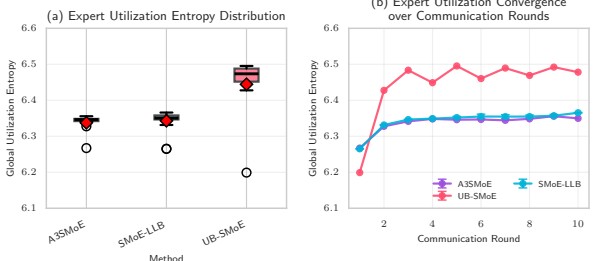

*Figure 2.* Global expert utilization entropy comparison. Higher entropy indicates more balanced utilization.

### 6.3.4. PSEUDO-GRADIENT ALIGNMENT FOR INACTIVE EXPERTS

To directly test whether PG provides a meaningful signal for experts that are inactive on low-resource clients, we measure cosine similarity between the server-side pseudo-gradient and the corresponding client-side true gradient on inactive experts.

Table 8 shows that the alignment is weak in shallow layers but becomes substantially positive in deeper layers, with a concatenated cosine similarity of 0.290. This trend is consistent with the observation that lower transformer FFN layers capture shallow or client-specific patterns, while upper layers encode more semantic, task-level information (Geva et al., 2021). Thus, PG is most reliable in the layers where cross-client task-level transfer is expected to be most useful, and it is strictly more informative than the zero-gradient update received by inactive experts in naive SMoE.

*Table 8.* Alignment between server pseudo-gradients and held-out client true gradients on inactive experts.

| Layer | Cos. sim. |
|---|---|
| 2 | 0.002 |
| 8 | 0.104 |
| 12 | 0.097 |
| 15 | 0.322 |
| Concat. | 0.290 |

## 7. Related Work

Federated fine-tuning methods for FMs include homogeneous approaches assuming uniform client resources (Chen et al., 2024; Xiong et al., 2025; Guo et al., 2023) and heterogeneous methods addressing resource diversity through logit distillation (Ma et al., 2024), personalized prompt generation (Yang et al., 2023), or adaptive LoRA ranks (Cho et al., 2024; Wang et al., 2024b; Bai et al., 2024; Ramesh & Dass, 2025). Recently, A$^3$SMoE (Tran et al., 2025) introduced heterogeneous sparsity by allowing client-specific expert activation, but suffers from expert utilization imbalance and gradient sparsity due to non-differentiable Top-K routing. Our work addresses these challenges directly.

**Distinction from Centralized Methods.** Loss-Free Balancing (Wang et al., 2024a) uses local batch statistics unsuitable for heterogeneous FL with non-IID data; our DMR uses global utilization. Default MoE (Panda et al., 2025) requires all experts available within a forward pass, which is infeasible in low-resource clients; our PG enables learning signals for experts clients never activate via cross-round information. More related works can be found in Appendix C.

## 8. Conclusion

We presented UB-SMoE, a resource-adaptive federated fine-tuning method that addresses expert utilization imbalance and non-differentiable Top-K routing in heterogeneous SMoE systems. Our convergence analysis reveals that these discordances induce an irreducible bias error floor scaling inversely with client computational budgets. UB-SMoE mitigates this through Dynamic Modulated Routing (DMR) and Universal Pseudo-Gradient (PG), which together form a self-reinforcing cycle that maintains expert viability across heterogeneous clients.

## Impact Statement

This work enables resource-constrained devices to participate in collaborative fine-tuning of foundation models, potentially democratizing access to AI capabilities. By reducing computational requirements for low-resource clients without sacrificing model quality, UB-SMoE could enable edge devices and organizations with limited resources to benefit from federated learning. As with all federated learning systems, privacy considerations remain important. While raw data is not shared, gradient information may leak sensitive information, and practitioners should consider privacy-preserving techniques (such as Differential Privacy) mechanisms when deploying in extreme sensitive domains.

## Acknowledgement

This publication has emanated from research conducted with the financial support of Research Ireland under Grant number 18/CRT/6222. This work was also supported by 6G-XCEL Project, Horizon Europe SNS Grant. Project code 214670/18401.

We thank Dr. Viet Quoc Pham (Trinity College Dublin, Ireland), Dr. Minh-Duong Nguyen (VinUniversity, Vietnam), and Khiem Le (University of Notre Dame, US) for the meaningful discussions at the initial phase of this project. We specifically thank Prof. Dinh-Thuc Nguyen (Ho Chi Minh City University of Science, Vietnam) for his support on the theoretical parts of this work.

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

## Appendix Overview

## A. Foundational Definitions and Notation

### A.1. Federated Fine-tuning with LoRA

**Definition 8.** (**Low-Rank Adaptation (LoRA)** (Hu et al., 2022)) For a pre-trained weight matrix $\mathbf{W}_0 \in \mathbb{R}^{d \times l}$, LoRA adapts this weight via a low-rank update parameterized by two trainable matrices, $B \in \mathbb{R}^{d \times r}$ and $A \in \mathbb{R}^{r \times l}$. We denote the set of trainable parameters as $\Theta = \{B, A\}$. The resulting adapted weight matrix $W(\Theta)$ is defined as: $W(\Theta) = W_0 + \Delta W(\Theta) = W_0 + \frac{\alpha}{r} BA$, where $r \in \mathbb{Z}^+ \ll \min(d, l)$ is the rank and $\alpha \in \mathbb{R}^+$ is the scaling factor. During fine-tuning, $W_0$ remains frozen, and optimization occurs only over $\Theta$.

**Definition 9.** (**Federated Fine-tuning with Homogeneous LoRA**) Let $\mathcal{X} \times \mathcal{Y}$ denote the input-output space. Consider $C$ clients, with $c \in [C] = \{1, \ldots, C\}$. Each client $c$ has local dataset $\mathcal{D}_c \subseteq \mathcal{X} \times \mathcal{Y}$. Assuming a homogeneous rank $r$ across all clients, we seek a shared set of global trainable parameters $\Theta$. Each client $c$'s local objective function is defined as: $F_c(\Theta) = \frac{1}{|\mathcal{D}_c|} \sum_{(x,y) \in \mathcal{D}_c} \mathcal{L}(W(\Theta); x, y)$, where $\mathcal{L}$ is the loss function and $W(\Theta)$ denotes the model weights adapted by $\Theta$ according to Definition 8. Federated fine-tuning with homogeneous LoRA solves the optimization problem:

$$\min_{\Theta} F(\Theta) = \sum_{c=1}^{C} p_c F_c(\Theta), \tag{17}$$

where $p_c = |\mathcal{D}_c| / \sum_{c'} |\mathcal{D}_{c'}|$ is the statistical weight of client $c$.

### A.2. Notation

We analyze the convergence over the space of trainable LoRA parameters $\Theta$. Let $\Theta^t$ be the global parameters at communication round $t$. $\Theta_c^{t,k}$ denotes the local parameters of client $c$ at local iteration $k$ ($k = 0, \ldots, \Gamma - 1$), initialized with $\Theta_c^{t,0} = \Theta^t$ and $\Gamma$ is the total number of local iterations per round. We utilize a decaying learning rate schedule, as specified in Theorem 4.1. Let $\eta_t$ be the effective global learning rate at round $t$, and $\eta_{l,t} = \eta_t / \Gamma$ be the corresponding local learning rate. $\Delta_t = F(\Theta^t) - F(\Theta^*)$ is the optimality gap, where $\Theta^*$ is a global optimum.

$U^t$ denotes the realized aggregated stochastic update used to update the global model: $\Theta^{t+1} = \Theta^t - \eta_t U^t$. $G^t = \mathbb{E}[U^t]$ is the expected aggregated update direction. $\mathbb{E}_t[\cdot]$ denotes the expectation conditioned on $\Theta^t$, while $g_c(\Theta)$ denotes the stochastic sparse gradient estimator used by client $c$. The expected sparse gradient evaluated at the global model is $\overline{g}(\Theta^t) = \sum_c p_c \mathbb{E}[g_c(\Theta^t)]$. Let $B(\Theta)$ represent the aggregate bias function, defined as $B(\Theta) = \nabla F(\Theta) - \overline{g}(\Theta)$ and $L_B$ is the Lipschitz constant of the bias (Assumption 5).

## B. Proofs

### B.1. Key Lemmas

We introduce two lemmas that bound the errors introduced by local updates (client drift) and the SMoE bias.

**Lemma 1.** (***Bounded Client Drift***). *Under Assumptions 1, 3, and 5, the expected aggregated update direction $G^t = \mathbb{E}[U^t]$ (where $U^t$ is the realized aggregated stochastic update) can be decomposed as:*

$$G^t = \nabla F(\Theta^t) - B(\Theta^t) + E_{drift}^t. \tag{18}$$

*Table 9.* Notation Summarization

| Notation | Description | Notation | Description |
|---|---|---|---|
| $p_c$ | Statistical weight of client $c$ | $W_0$ | Pre-trained weight matrix. |
| $\beta_c$ | Computational budget of client $c$, $\beta_c \in [0,1]$. | $B, A$ | Trainable LoRA matrices. |
| $T, \Gamma$ | Number of communication rounds and local iterations per round. | $M$ | Total number of experts in an SMoE layer. |
| $\eta_t, \eta_{l,t}$ | Global and local learning rates at round $t$. | $W_i^{(e)}, W^{(r)}$ | Weights of the $i$-th expert and the router. |
| $\Theta^*$ | Global optimum. | $K_{max}$ | Maximum allowable sparsity. |
| $\Theta^t, \Theta_c^{t,k}$ | Global parameters at round $t$ and local parameters at iteration $k$. | $K_c$ | Client-specific sparsity level, $K_c = \lfloor K_{max}\beta_c \rfloor$. |
| $\Theta_i^{(e)}, \Theta^{(r)}$ | LoRA parameters for the $i$-th expert and the router. | $\mathcal{A}_c(x)$ | Activation set (indices of the Top-$K_c$ experts) for client $c$. |
| $B(\Theta)$ | Aggregate bias of the gradient estimator. | $\gamma_i(s)$ | Gating weight for expert $i$. |
| $g_{c,i}$ | Sparse stochastic gradient. | $\mathbb{I}(\cdot)$ | Indicator function. |
| $\Delta_t$ | Optimality gap, $F(\Theta^t) - F(\Theta^*)$. | $L_B$ | Lipschitz constant of the bias function. |
| $E_{drift}^t$ | Client drift error at round $t$. | $L_{eff}$ | Effective Lipschitz constant ($L + L_B$). |

*The client drift error $E_{drift}^t$ is bounded by:*

$$\mathbb{E}[\|E_{drift}^t\|^2] \leq C_D \eta_t^2 G^2, \tag{19}$$

*where $C_D = L_{eff}^2 \frac{(\Gamma-1)(2\Gamma-1)}{6\Gamma^2}$ is a constant depending on $L_{eff} = L + L_B$ (Assumption 5) and $\Gamma$.*

*Proof.* The drift error is defined as the difference between the expected aggregated update $G^t$ and the expected sparse gradient evaluated at the global model $\bar{g}(\Theta^t) = \sum_c p_c \mathbb{E}[g_c(\Theta^t)]$:

$$E_{drift}^t = G^t - \bar{g}(\Theta^t) = \sum_c p_c \frac{1}{\Gamma} \sum_{k=0}^{\Gamma-1} \left( \mathbb{E}[g_c(\Theta_c^{t,k})] - \mathbb{E}[g_c(\Theta^t)] \right). \tag{20}$$

We analyze the Lipschitz continuity of the expected sparse gradient $\mathbb{E}[g_c(\Theta)] = \nabla F_c(\Theta) - B_c(\Theta)$. By Assumption 1 and 5:

$$\|\mathbb{E}[g_c(\Theta_1)] - \mathbb{E}[g_c(\Theta_2)]\| \leq (L + L_B)\|\Theta_1 - \Theta_2\| = L_{eff}\|\Theta_1 - \Theta_2\|. \tag{21}$$

We bound the squared norm, $\mathbb{E}[\|E_{drift}^t\|^2]$. By applying Jensen's inequality:

$$\mathbb{E}[\|E_{drift}^t\|^2] \leq \sum_c p_c \frac{1}{\Gamma} \sum_{k=0}^{\Gamma-1} \mathbb{E}[\|\mathbb{E}[g_c(\Theta_c^{t,k})] - \mathbb{E}[g_c(\Theta^t)]\|^2]$$
$$\leq L_{eff}^2 \sum_c p_c \frac{1}{\Gamma} \sum_{k=0}^{\Gamma-1} \mathbb{E}[\|\Theta_c^{t,k} - \Theta^t\|^2] \tag{22}$$

We bound the expected divergence of the local model $\Theta_c^{t,k}$ from the initial global model $\Theta^t$. The local model at step $k$ is $\Theta_c^{t,k} = \Theta^t - \eta_{l,t} \sum_{j=0}^{k-1} g_c(\Theta_c^{t,j})$.

$$\mathbb{E}[\|\Theta_c^{t,k} - \Theta^t\|^2] = \eta_{l,t}^2 \mathbb{E}\left[ \left\| \sum_{j=0}^{k-1} g_c(\Theta_c^{t,j}) \right\|^2 \right]. \tag{23}$$

Using Cauchy-Schwarz ($|| \sum v_j ||^2 \leq k \sum ||v_j||^2$) and Assumption 3:

$$
\begin{aligned}
\mathbb{E}[||\Theta_c^{t,k} - \Theta^t||^2] &\leq \eta_{l,t}^2 k \sum_{j=0}^{k-1} \mathbb{E}[||g_c(\Theta_c^{t,j})||^2] \\
&\leq \eta_{l,t}^2 k^2 G^2.
\end{aligned}
\tag{24}
$$

Substituting this bound back into Eq.22:

$$
\mathbb{E}[||E_{drift}^t||^2] \leq \frac{L_{eff}^2 \eta_{l,t}^2 G^2}{\Gamma} \sum_{k=0}^{\Gamma-1} k^2.
\tag{25}
$$

Using the formula $\sum_{k=0}^{\Gamma-1} k^2 = \frac{(\Gamma-1)\Gamma(2\Gamma-1)}{6}$ and substituting $\eta_{l,t} = \eta_t / \Gamma$:

$$
\begin{aligned}
\mathbb{E}[||E_{drift}^t||^2] &\leq L_{eff}^2 (\frac{\eta_t}{\Gamma})^2 G^2 \frac{(\Gamma-1)(2\Gamma-1)}{6} \\
&= \left( L_{eff}^2 \frac{(\Gamma-1)(2\Gamma-1)}{6\Gamma^2} \right) \eta_t^2 G^2 = C_D \eta_t^2 G^2.
\end{aligned}
\tag{26}
$$

$\square$

**Lemma 2.** *(**Bias Evolution Bound**). Under Assumptions 2 and 5, the squared norm of the aggregate bias at $W^t$ can be bounded as:*

$$
||B(\Theta^t)||^2 \leq 2||B(\Theta^*)||^2 + \frac{4L_B^2}{\mu}\Delta_t.
\tag{27}
$$

*Proof.* We add and subtract the bias at the optimal point, $B(\Theta^*)$, inside the norm:

$$
\begin{aligned}
||B(\Theta^t)||^2 &= ||(B(\Theta^t) - B(\Theta^*)) + B(\Theta^*)||^2 \\
&\leq 2||B(\Theta^t) - B(\Theta^*)||^2 + 2||B(\Theta^*)||^2
\end{aligned}
\tag{28}
$$

Using Assumption 5: $||B(\Theta^t) - B(\Theta^*)||^2 \leq L_B^2 ||\Theta^t - \Theta^*||^2$, the Eq. 28 becomes:

$$
||B(\Theta^t)||^2 \leq 2(L_B^2 ||\Theta^t - \Theta^*||^2) + 2||B(\Theta^*)||^2
\tag{29}
$$

Now, we utilize Assumption 2, which is established in optimization theory (Karimi et al., 2016) that the PL condition implies the Quadratic Growth (QG) condition. Let $\mathcal{S}^*$ be the set of global optima and $F^*$ be the optimal value. The QG condition states:

$$
F(\Theta^t) - F^* \geq \frac{\mu}{2}||\Theta^t - \text{proj}_{\mathcal{S}^*}(\Theta^t)||^2,
\tag{30}
$$

where $\text{proj}_{\mathcal{S}^*}(\Theta^t)$ is the projection of $\Theta^t$ onto $\mathcal{S}^*$. For the remainder of this proof, we let $\Theta^*$ denote this specific projection. Rearranging the QG condition provides the required bound on the parameter distance:

$$
||\Theta^t - \Theta^*||^2 \leq \frac{2}{\mu}(F(\Theta^t) - F(\Theta^*))
\tag{31}
$$

$$
\leq \frac{2}{\mu}\Delta_t \quad \text{(with } \Delta_t = F(\Theta^t) - F(\Theta^*))
\tag{32}
$$

Finally, Equation 29 becomes:

$$
\begin{aligned}
||B(\Theta^t)||^2 &\leq 2L_B^2 \left( \frac{2}{\mu}\Delta_t \right) + 2||B(\Theta^*)||^2 \\
&= \frac{4L_B^2}{\mu}\Delta_t + 2||B(\Theta^*)||^2
\end{aligned}
\tag{33}
$$

$\square$

**Lemma 3.** *(**Square-Root Exponentially Weighted Sum Bound**). Let $c > 0$, the following inequality holds for $T \geq 2$:*

$$\sum_{k=1}^{T} \frac{1}{k} \exp(-c(\sqrt{T} - \sqrt{k})) \leq \frac{4}{c\sqrt{T}} + \mathcal{O}\left(\frac{1}{T}\right). \tag{34}$$

*Proof.* Indeed, splitting the summation into two parts: $1 \leq k \leq T/2$ and $T/2 < k \leq T$. Let $g(x) = \exp(-c(\sqrt{T} - \sqrt{x}))$. In the range $1 \leq k \leq T/2$, we have $\sqrt{T} - \sqrt{k} \geq \sqrt{T} - \sqrt{T/2} = \sqrt{T}(1 - 1/\sqrt{2})$. Let $c' = c(1 - 1/\sqrt{2}) > 0$:

$$\sum_{k=1}^{\lfloor T/2 \rfloor} \frac{1}{k} g(k) \leq \exp(-c'\sqrt{T}) \sum_{k=1}^{\lfloor T/2 \rfloor} \frac{1}{k} = \mathcal{O}(\exp(-c'\sqrt{T}) \log T). \tag{35}$$

This term decays exponentially fast.

In the range $T/2 < k \leq T$, we use the bound $1/k \leq 2/T$:

$$\sum_{k=\lfloor T/2 \rfloor + 1}^{T} \frac{1}{k} g(k) \leq \frac{2}{T} \sum_{k=\lfloor T/2 \rfloor + 1}^{T} g(k). \tag{36}$$

Since $g(x)$ is an increasing function, we bound the sum using an integral approximation:

$$\sum_{k=\lfloor T/2 \rfloor + 1}^{T} g(k) \leq \int_{\lfloor T/2 \rfloor + 1}^{T} g(x)dx + g(T) \leq \int_{T/2}^{T} \exp(-c(\sqrt{T} - \sqrt{x}))dx + 1. \tag{37}$$

We analyze the integral $I = \int_{T/2}^{T} \exp(-c(\sqrt{T} - \sqrt{x}))dx$. Let $v = \sqrt{T} - \sqrt{x}$, then $x = (\sqrt{T} - v)^2$ and $dx = -2(\sqrt{T} - v)dv$. The limits change from $x = T/2$ (where $v = v_{max} = \sqrt{T}(1 - 1/\sqrt{2})$) to $x = T$ (where $v = 0$).

$$I = \int_{0}^{v_{max}} \exp(-cv)2(\sqrt{T} - v)dv = 2\sqrt{T} \int_{0}^{v_{max}} \exp(-cv)dv - 2\int_{0}^{v_{max}} v\exp(-cv)dv, \tag{38}$$

We evaluate the integrals:

$$\int_{0}^{v_{max}} \exp(-cv)dv = \frac{1}{c}(1 - \exp(-cv_{max})), \quad \int_{0}^{v_{max}} v\exp(-cv)dv = \frac{1}{c^2}(1 - \exp(-cv_{max})(1 + cv_{max})) \tag{39}$$

Since $v_{max} = \Theta(\sqrt{T})$, the terms involving $\exp(-cv_{max})$ decay exponentially:

$$\begin{aligned} I &= \frac{2\sqrt{T}}{c}(1 - e^{-cv_{max}}) - \frac{2}{c^2}(1 - e^{-cv_{max}}(1 + cv_{max})) \\ &= \frac{2\sqrt{T}}{c} - \frac{2}{c^2} + \mathcal{O}(\sqrt{T}e^{-\Omega(\sqrt{T})}). \end{aligned} \tag{40}$$

Substituting this bound back into Eq. 37:

$$\sum_{k=\lfloor T/2 \rfloor + 1}^{T} \frac{1}{k} g(k) \leq \frac{2}{T}(I + 1) = \frac{2}{T}\left(\frac{2\sqrt{T}}{c} - \frac{2}{c^2} + 1 + \ldots\right) = \frac{4}{c\sqrt{T}} + \mathcal{O}\left(\frac{1}{T}\right). \tag{41}$$

$\square$

## B.2. Proof of Theorem 4.1

*Proof.* We begin by analyzing the expected progress after one communication round. Let $\mathbb{E}_t[\cdot]$ denote the expectation conditioned on the parameters $\Theta^t$ at the start of round $t$. By Assumption 1:

$$\mathbb{E}[F(\Theta^{t+1})] \leq \mathbb{E}[F(\Theta^t) + \langle \nabla F(\Theta^t), \Theta^{t+1} - \Theta^t \rangle + \frac{L}{2}||\Theta^{t+1} - \Theta^t||^2] \tag{42}$$

The global update is $\Theta^{t+1} - \Theta^t = -\eta_t U^t$. Substituting this and subtracting the optimal value $F(\Theta^*)$ from both sides yields the recurrence for the optimality gap $\Delta_t$:

$$\Delta_{t+1} \leq \Delta_t - \eta_t \langle \nabla F(\Theta^t), U^t \rangle + \frac{L\eta_t^2}{2}||U^t||^2 \tag{43}$$

By Assumption 3, we assume $\mathbb{E}_t[||U^t||^2] \leq G^2$. Since $\nabla F(\Theta^t)$ is fixed given $\Theta^t$, we have $\mathbb{E}_t[\langle \nabla F(\Theta^t), U^t \rangle] = \langle \nabla F(\Theta^t), \mathbb{E}_t[U^t] \rangle$. Let $G^t = \mathbb{E}_t[U^t]$:

$$\mathbb{E}_t[\Delta_{t+1}] \leq \Delta_t - \eta_t \langle \nabla F(\Theta^t), G^t \rangle + \frac{L\eta_t^2 G^2}{2}. \tag{44}$$

We substitute the decomposition of $G^t$ from Lemma 1, $G^t = \nabla F(\Theta^t) - B(\Theta^t) + E_{drift}^t$, we have:

$$\begin{aligned}
\mathbb{E}_t[\Delta_{t+1}] &\leq \Delta_t - \eta_t \langle \nabla F(\Theta^t), \nabla F(\Theta^t) - B(\Theta^t) + E_{drift}^t \rangle + \frac{L\eta_t^2 G^2}{2} \\
&= \Delta_t - \eta_t ||\nabla F(\Theta^t)||^2 + \eta_t \langle \nabla F(\Theta^t), B(\Theta^t) \rangle - \eta_t \langle \nabla F(\Theta^t), E_{drift}^t \rangle + \frac{L\eta_t^2 G^2}{2}.
\end{aligned} \tag{45}$$

We utilize Young's inequality, which states that for any vectors $u, v$ and scalar $\alpha > 0$, we have $|\langle u, v \rangle| \leq \frac{\alpha}{2}||u||^2 + \frac{1}{2\alpha}||v||^2$. We choose $\alpha = 1/2$. First, we bound the bias term:

$$\langle \nabla F(\Theta^t), B(\Theta^t) \rangle \leq \frac{1}{4}||\nabla F(\Theta^t)||^2 + ||B(\Theta^t)||^2$$

Next, we bound the drift term:

$$-\langle \nabla F(\Theta^t), E_{drift}^t \rangle \leq |\langle \nabla F(\Theta^t), E_{drift}^t \rangle| \leq \frac{1}{4}||\nabla F(\Theta^t)||^2 + ||E_{drift}^t||^2$$

Therefore Eq.45 becomes:

$$\begin{aligned}
\mathbb{E}[\Delta_{t+1}] &\leq \mathbb{E}[\Delta_t] - \eta_t \mathbb{E}[||\nabla F(\Theta^t)||^2] + \eta_t \mathbb{E}[\frac{1}{4}||\nabla F(\Theta^t)||^2 + ||B(\Theta^t)||^2] \\
&\quad + \eta_t \mathbb{E}[\frac{1}{4}||\nabla F(\Theta^t)||^2 + ||E_{drift}^t||^2] + \frac{L\eta_t^2 G^2}{2} \\
&= \mathbb{E}[\Delta_t] - \frac{\eta_t}{2}\mathbb{E}[||\nabla F(\Theta^t)||^2] + \eta_t \mathbb{E}[||B(\Theta^t)||^2] + \eta_t \mathbb{E}[||E_{drift}^t||^2] + \frac{L\eta_t^2 G^2}{2} \\
&= \mathbb{E}[\Delta_t] - \eta_t \mu \mathbb{E}[\Delta_t] + \eta_t \mathbb{E}[||B(\Theta^t)||^2] + \eta_t \mathbb{E}[||E_{drift}^t||^2] + \frac{L\eta_t^2 G^2}{2} \quad \text{(By Assumption 2)} \\
&= \mathbb{E}[\Delta_t] - \eta_t \mu \mathbb{E}[\Delta_t] + \frac{4\eta_t L_B^2}{\mu}\mathbb{E}[\Delta_t] + 2\eta_t ||B(\Theta^*)||^2 + \eta_t \mathbb{E}[||E_{drift}^t||^2] + \frac{L\eta_t^2 G^2}{2} \quad \text{(By Lemma 2)} \\
&= \mathbb{E}[\Delta_t] - \eta_t \mu \mathbb{E}[\Delta_t] + \frac{4\eta_t L_B^2}{\mu}\mathbb{E}[\Delta_t] + 2\eta_t ||B(\Theta^*)||^2 + C_D \eta_t^3 G^2 + \frac{L\eta_t^2 G^2}{2} \quad \text{(By Lemma 1)} \\
&= \left(1 - \eta_t \mu + \frac{4\eta_t L_B^2}{\mu}\right)\mathbb{E}[\Delta_t] + 2\eta_t ||B(\Theta^*)||^2 + C_D \eta_t^3 G^2 + \frac{L\eta_t^2 G^2}{2}
\end{aligned} \tag{46}$$

Let $C_B = 2\|B(\Theta^*)\|^2$, $\mu' = \mu - \frac{4L_B^2}{\mu}$ denotes the bias-penalized curvature, and $V_t = C_D G^2 \eta_t^3 + \frac{LG^2}{2}\eta_t^2$ presents the variance and drift term. The recurrence is:

$$\mathbb{E}[\Delta_{t+1}] \leq (1 - \eta_t \mu')\mathbb{E}[\Delta_t] + \eta_t C_B + V_t. \tag{47}$$

We use the learning rate $\eta_t = \frac{1}{\sqrt{\Gamma(t+1)}}$ and let $\gamma = \sqrt{\Gamma}$. Assuming $\Gamma$ is large enough such that $\eta_t \mu' \leq 1$. Unrolling the recursion from $t = 0$ to $T - 1$:

$$
\begin{aligned}
\mathbb{E}[\Delta_T] &\leq \Phi_{T-1,0}\mathbb{E}[\Delta_0] + \sum_{t=0}^{T-1} \Phi_{T-1,t+1}(\eta_t C_B + V_t) \\
&= \underbrace{\Phi_{T-1,0}\mathbb{E}[\Delta_0]}_{(A1)} + C_B \underbrace{\sum_{t=0}^{T-1} \Phi_{T-1,t+1}\eta_t}_{(A2)} + \underbrace{\sum_{t=0}^{T-1} \Phi_{T-1,t+1}V_t}_{(A3)},
\end{aligned}
\tag{48}
$$

where $\Phi_{k,j} = \prod_{i=j}^{k}(1 - \eta_i \mu')$ is the contraction factor.

We analyze each term:

- **Term (A1)**: Using $1 - x \leq \exp(-x)$, $\Phi_{T-1,0} \leq \exp(-\mu' \sum_{t=0}^{T-1} \eta_t)$. Since:

$$\sum_{t=0}^{T-1} \eta_t = \frac{1}{\gamma}\sum_{k=1}^{T} \frac{1}{\sqrt{k}} \geq \frac{1}{\gamma}\int_1^{T+1} \frac{1}{\sqrt{x}}dx = \frac{2}{\gamma}(\sqrt{T+1} - 1),$$

  the term (A1) decays exponentially fast with $\sqrt{T}$.

- **Term (A2)**: We utilize the identity $\sum_{t=0}^{T-1} \Phi_{T-1,t+1}(\eta_t \mu') = 1 - \Phi_{T-1,0}$:

$$S_B = \frac{C_B}{\mu'}(1 - \Phi_{T-1,0}).$$

  As $T \to \infty$, this converges to the bias error floor $B_{SMoE} = \frac{C_B}{\mu'} = \frac{2\|B(\Theta^*)\|^2}{\mu'}$.

- **Term (A3)**: We need to bound $V_t$:

$$V_t = \frac{G^2}{\gamma^2(t+1)}\left(\frac{C_D}{\gamma\sqrt{t+1}} + \frac{L}{2}\right) \leq \frac{C_V}{t+1}, \tag{49}$$

  where $C_V = \frac{G^2}{\gamma^2}(\frac{C_D}{\gamma} + \frac{L}{2})$ (with $\gamma \geq 1, t \geq 0$).

  We need a lower bound on the sum of learning rates to bound the contraction factor. Using integral approximation for decreasing functions:

$$\sum_{j=t+1}^{T-1} \eta_j = \frac{1}{\gamma}\sum_{k=t+2}^{T} \frac{1}{\sqrt{k}} \geq \frac{1}{\gamma}\int_{t+2}^{T+1} \frac{1}{\sqrt{x}}dx = \frac{2}{\gamma}(\sqrt{T+1} - \sqrt{t+2}). \tag{50}$$

  Let $c' = \frac{2\mu'}{\gamma}$, then $\Phi_{T-1,t+1} \leq \exp(-c'(\sqrt{T+1} - \sqrt{t+2}))$. So:

$$
\begin{aligned}
(A3) &\leq C_V \sum_{t=0}^{T-1} \frac{1}{t+1}e^{-c'(\sqrt{T+1}-\sqrt{t+2})} \\
&= C_V \sum_{k=2}^{T+1} \frac{1}{k-1}e^{-c'(\sqrt{T'}-\sqrt{k})}. \quad \text{(Let } T' = T+1 \text{ and} k = t+1)
\end{aligned}
\tag{51}
$$

Since $\frac{1}{k-1} \leq \frac{2}{k}$ for $k \geq 2$, we have:

$$(A3) \leq 2C_V \sum_{k=2}^{T+1} \frac{1}{k} e^{-c'(\sqrt{T'}-\sqrt{k})}. \tag{52}$$

We apply Lemma 3 to this summation:

$$(A3) \leq 2C_V \left( \frac{4}{c'\sqrt{T+1}} + \mathcal{O}\left(\frac{1}{T}\right) \right) = \frac{8C_V}{c'\sqrt{T+1}} + \mathcal{O}\left(\frac{1}{T}\right). \tag{53}$$

Substituting the constants $c' = \frac{2\mu'}{\gamma}, \gamma = \sqrt{\Gamma}$ and $C_V = \frac{G^2}{\gamma^2}\left(\frac{C_D}{\gamma} + \frac{L}{2}\right)$, the Eq. 53 becomes:

$$\begin{aligned}
(A3) &\leq \frac{8C_V}{c'\sqrt{T+1}} = \frac{8 \cdot \frac{G^2}{\gamma^2}\left(\frac{C_D}{\gamma} + \frac{L}{2}\right)}{\frac{2\mu'}{\gamma}\sqrt{T+1}} = \frac{4G^2}{\gamma\mu'\sqrt{T+1}}\left(\frac{C_D}{\gamma} + \frac{L}{2}\right) \\
&= \frac{G^2}{\mu'\sqrt{\Gamma(T+1)}}\left(\frac{4C_D}{\sqrt{\Gamma}} + 2L\right) \\
&= \mathcal{O}\left(\frac{G^2(C_D + L)}{\mu'\sqrt{\Gamma T}}\right)
\end{aligned} \tag{54}$$

Combining all terms, Eq. 48 becomes:

$$\mathbb{E}[F(\Theta^T)] - F(\Theta^*) \leq \mathcal{O}\left(\frac{G^2(C_D + L)}{\mu'\sqrt{\Gamma T}}\right) + \mathcal{O}\left(\frac{||B(\Theta^*)||^2}{\mu'}\right). \tag{55}$$

From Lemma 1, we have: $C_D = L_{eff}^2 \frac{(\Gamma-1)(2\Gamma-1)}{6\Gamma^2}$. If we treat the number of local steps $\Gamma$ as a constant, then $C_D$ is bounded by a constant proportional to $L_{eff}^2$, we get:

$$\mathbb{E}[F(\Theta^T)] - F(\Theta^*) \leq \underbrace{\mathcal{O}\left(\frac{G^2(L + L_{eff}^2)}{\mu'\sqrt{\Gamma T}}\right)}_{\text{Optimization Error}} + \underbrace{B_{SMoE}}_{\text{Bias Error}}. \tag{56}$$

$\square$

## C. Detailed Related Work

### C.1. Parameter-Efficient Federated Fine-Tuning for Foundation Models

Federated fine-tuning approaches for FMs can be classified into two primary categories: (i) Homogeneous federated fine-tuning and (ii) Heterogeneous federated fine-tuning.

Homogeneous methods assume uniform client resources, which means clients have similar computational resources for fine-tuning FMs. FedDAT (Chen et al., 2024) introduces a dual-adapter approach, which consists of a frozen copy of the global adapter to retain general knowledge and a local adapter to learn client-specific features. To fine-tune on different tasks across clients, Pilot (Xiong et al., 2025) proposes a two-stage adapter framework that first uses task-specific and client-specific adapters to extract distinct features and then builds a cross-task adapter to learn general knowledge. Prompt tuning is another fine-tuning technique in which a small set of soft prompts is prepended to input embeddings, effectively reducing communication between clients. PromptFL (Pan et al., 2024) trains shared soft prompts on FMs while pFedPrompt (Guo et al., 2023) personalizes soft prompts for each client. FedIT (Zhang et al., 2024) enables each client to fine-tune LoRA modules using their local data while freezing almost all parameters.

Heterogeneous methods focus on addressing the system heterogeneity challenge where clients have different computational capabilities. FedHPL (Ma et al., 2024) employs a global logit distillation scheme that allows the server to aggregate

model-agnostic output logits from clients. pFedPG (Yang et al., 2023) learns a prompt generator on the server via clients' optimization directions that produces personalized prompts for each client. FedSelect (Tamirisa et al., 2023) proposes a gradient-based lottery ticket method that selects the most impactful parameters for local fine-tuning; thus, each client can personalize an optimal subnetwork. Instead of allocating fixed ranks to clients like FedIT, HetLoRA (Cho et al., 2024) employs a rank self-pruning mechanism that allows different clients to use different ranks. Other works propose a novel aggregation that inherently supports the use of heterogeneous LoRA-ranks across different clients by designing a stacking-based aggregation (Wang et al., 2024b) or SVD-based weight redistribution (Bai et al., 2024). FLoRIST (Ramesh & Dass, 2025) improves upon FlexLoRA's aggregation by performing a fast SVD directly on the clients' stacked low-rank adapters to operate within a more efficient, compact space.

Recently, A$^3$SMoE (Tran et al., 2025) extends the conditional computation property of SMoE which allows each client to activate a variable number of experts based on its available resources. However, this method faces two limitations: (i) expert utilization imbalance, and (ii) non-differentiability of Top-K routing, failing to achieve optimal performance. In this work, we propose the UB-SMoE method to address these two challenges via Dynamic Modulated Routing and Universal Pseudo-Gradient mechanisms.

### C.2. Sparse Mixture-of-Experts (SMoE)

The concept of SMoE is introduced by Shazeer et al. (Shazeer et al., 2017), which enables scaling neural networks by partitioning model capacity across multiple specialized experts while activating only a subset for each input. Unlike dense MoE, SMoE employs a Top-K routing mechanism that selects a small subset of experts per token, thereby decoupling model capacity from inference cost. This sparse activation pattern has enabled models such as Switch Transformer (Fedus et al., 2022), GLaM (Du et al., 2022), Mixtral (Jiang et al., 2024), and DeepSeek-MoE (Liu et al., 2024a) to scale to hundreds of billions of parameters at manageable computational budgets.

Despite their computational advantages, training SMoE models face several challenges. The most prominent issue is *load imbalance*, where the router learns to preferentially route tokens to a small subset of experts, causing routing collapse (Shazeer et al., 2017) or increased computational overhead (Fedus et al., 2022; Shazeer et al., 2017). To mitigate load imbalance, Shazeer et al. (Shazeer et al., 2017) introduces an auxiliary load balancing loss that penalize uneven token distribution across experts. This approach is simplified by Fedus et al. (Fedus et al., 2022) to encourage the product of the fraction of tokens routed to each expert and the average router probability to be uniform. GShard (Lepikhin et al., 2021) develops a novel gating function to limit the maximum number of tokens each expert can process, with overflow tokens either dropped or passed to subsequent layers. ST-MoE (Zoph et al., 2022) contributes the router z-loss, which penalizes large logits in the gating mechanism to improve training stability without degrading model quality. Qiu et al. (Qiu et al., 2025) addresses the load imbalance issue at the global-batch level by synchronizing expert selection frequencies across parallel groups to calculate the load-balancing loss over the entire global batch.

However, one crucial limitation of these auxiliary losses is that they introduce unexpected gradients that conflict with the model objective. Recent work has explored auxiliary-loss-free approaches to load balancing. Wang et al. (Wang et al., 2024a) proposes Loss-Free Balancing, which applies expert-wise biases to routing scores that are dynamically updated based on recent expert load statistics. This approach avoids the interference gradients introduced by auxiliary losses while maintaining balanced token distribution. A theoretical framework is introduced by Han & Zhong (Han & Zhong, 2025) that provides convergence guarantees for this auxiliary-loss-free paradigm.

A second fundamental challenge of SMoE arises from the *non-differentiability of Top-K routing*. To address this issue, several gradient estimation techniques have been developed. SparseMixer (Liu et al., 2023) utilizes numerical ordinary differential equation solvers to approximate the gradient contribution from non-activated experts. SparseMixer-v2 (Liu et al., 2024b) improves this idea by utilizing Heun's third-order method for more accurate gradient approximation. ReMoE (Wang et al., 2025) replaces the standard TopK+Softmax routing with a continuous ReLU-based routing function. The continuous nature of ReLU enables the training pipeline to become fully differentiable, which means that the router can learn exactly when to activate or deactivate an expert via standard backpropagation. Recently, Default MoE (Panda et al., 2025) has introduced a lightweight approximation that allows the router to receive a dense gradient update (signals from all experts) while maintaining the computational efficiency of sparse activation. In this method, missing activations for each expert are computed as an exponential moving average of that expert's outputs from previous tokens that did activate it.

## C.3. Biased Optimization

Standard SGD typically assumes access to unbiased gradient estimators, where the expectation of the stochastic gradient equals the true gradient of the objective function (Bottou et al., 2018). However, real-world scenarios introduce biases into the gradient estimators, including gradient compression and quantization for communication efficiency (Li et al., 2019a), conditional stochastic optimization with nested expectations (Hu et al., 2021), and reinforcement learning with function approximation (Liu et al., 2022).

The theoretical foundations for SGD with biased gradients have been established. Karimi et al. (Karimi et al., 2019) establishes non-asymptotic convergence guarantees for stochastic approximation on non-convex smooth objectives. This framework characterizes the bias through an alignment condition between the update direction and the true gradient, proving convergence to a neighborhood of the stationary point determined by the bias magnitude. Ajalloeian & Stich (Ajalloeian & Stich, 2020) analyzes biased SGD under strongly convex, convex, and weakly convex objectives. The authors demonstrate that biased gradient methods can achieve near-optimal rates when the bias decays appropriately with increased sampling. Hu et al. (Hu et al., 2021) addresses conditional stochastic optimization by proposing biased stochastic gradient descent to handle gradient bias arising from estimating nested expectations. The authors also establish optimal sample complexity bounds for convex and non-convex objectives by explicitly controlling the bias-variance tradeoff through inner mini-batch sampling. Demidovich et al. (Demidovich et al., 2023) unifies the fragmented theoretical landscape by introducing a novel framework, which is proven to be the weakest and most general assumption encompassing prior models. Utilizing this framework, the authors demonstrate that biased SGD can maintain optimal convergence rates even under practical bias sources such as subsampling and compression.

In our convergence analysis, we leverage these theoretical frameworks to characterize the convergence behavior of federated SMoE fine-tuning.

# D. Pseudo-code

This section provides detailed pseudo-code for our proposed UB-SMoE method. We first present the overall federated fine-tuning framework (Algorithm 1), followed by the detailed procedures for Dynamic Modulated Routing (Algorithm 2) and Universal Pseudo-Gradient (Algorithm 3).

## D.1. Overall UB-SMoE Framework

Algorithm 1 presents the complete UB-SMoE procedure. Each round, the server broadcasts model parameters, modulation vectors $\phi$, and pseudo-gradient buffers $\tilde{\mathbf{G}}$ to clients. Clients perform local training using DMR for expert selection (with client-specific sparsity $K_c$) and inject pseudo-gradients for non-activated experts. Upon completion, clients upload updated parameters and activation statistics. The server aggregates via FedAvg, computes pseudo-gradients from parameter changes, and updates modulation parameters based on global utilization.

---

**Algorithm 1** UB-SMoE: Federated Fine-tuning with Dynamic Modulated Routing and Pseudo-Gradients

---

**Require:** Clients $C$, rounds $T$, local iterations $\Gamma$, budgets $\{\beta_c\}_{c=1}^C$, max sparsity $K_{\max}$, experts $M$, learning rate $\eta$, candidate size $N_p$, momentum $\zeta$
**Ensure:** Global model parameters $\Theta^{(T)}$
1: **Server Initialization:**
2: Initialize global LoRA parameters $\Theta^{(0)} = \{\Theta_i^{(e)}\}_{i=1}^M \cup \Theta^{(r)}$
3: Initialize modulation vectors $\boldsymbol{\phi}^{(l)} \leftarrow \mathbf{0} \in \mathbb{R}^M$ for each layer $l \in [L]$
4: Initialize pseudo-gradient buffers $\tilde{\mathbf{G}}^{(l)} \leftarrow \mathbf{0}$ for each layer $l \in [L]$
5: Compute average sparsity $\bar{K} \leftarrow \sum_{c=1}^C p_c \lfloor K_{\max} \cdot \beta_c \rfloor$
6: **for** round $t = 0, 1, \ldots, T-1$ **do**
7:    **Server broadcasts:** $\Theta^{(t)}, \{\boldsymbol{\phi}^{(l,t)}\}_{l=1}^L, \{\tilde{\mathbf{G}}^{(l,t)}\}_{l=1}^L$
8:    **for** each client $c \in [C]$ **in parallel do**
9:       $K_c \leftarrow \lfloor K_{\max} \cdot \beta_c \rfloor$;    $\rho_c \leftarrow \sqrt{\bar{K}/K_c}$             // Client sparsity and scaling
10:       $\Theta_c^{(t,0)} \leftarrow \Theta^{(t)}$;    $\mathbf{a}_c^{(l)} \leftarrow \mathbf{0}, n_c^{(l)} \leftarrow 0$ for all $l$
11:       **for** local iteration $k = 0, \ldots, \Gamma-1$ **do**
12:          Sample mini-batch $\mathcal{B}_c \sim \mathcal{D}_c$
13:          **for** each token $x \in \mathcal{B}_c$, each layer $l \in [L]$ **do**
14:             $\mathcal{A}_c(x), \boldsymbol{\gamma} \leftarrow \text{DMR}(x, W^{(r,l)}, \boldsymbol{\phi}^{(l,t)}, K_c, N_p)$          // Alg. 2
15:             $\mathbf{a}_c^{(l)} \leftarrow \mathbf{a}_c^{(l)} + \mathbf{1}_{\mathcal{A}_c(x)}$;    $n_c^{(l)} \leftarrow n_c^{(l)} + 1$
16:             $y^{(l)} \leftarrow \sum_{i \in \mathcal{A}_c(x)} \gamma_i \cdot \mathbb{E}_i(x; \Theta_i^{(e,l)})$
17:          **end for**
18:          $\mathcal{L}_c \leftarrow \mathcal{L}_{\text{task}}(\Theta_c^{(t,k)}; \mathcal{B}_c) + \lambda \sum_l \|\boldsymbol{\phi}^{(l)}\|_2^2$          // Eq. 11
19:          $\hat{\mathbf{g}}_c \leftarrow \text{PSEUDOGRAD}(\nabla_\Theta \mathcal{L}_c, \mathcal{A}_c, \tilde{\mathbf{G}}^{(t)}, \rho_c)$          // Alg. 3
20:          $\Theta_c^{(t,k+1)} \leftarrow \Theta_c^{(t,k)} - \eta \cdot \hat{\mathbf{g}}_c$
21:       **end for**
22:       **Client uploads:** $\Theta_c^{(t,\Gamma)}, \{(\mathbf{a}_c^{(l)}, n_c^{(l)})\}_{l=1}^L$
23:    **end for**
24:    **Server Aggregation:**
25:    $\Theta^{(t+1)} \leftarrow \sum_{c=1}^C p_c \cdot \Theta_c^{(t,\Gamma)}$             // FedAvg
26:    **for** each layer $l \in [L]$ **do**
27:       $\tilde{G}_i^{(l,t+1)} \leftarrow (\Theta_i^{(e,l,t)} - \Theta_i^{(e,l,t+1)})/(\eta\Gamma)$ for all $i \in [M]$          // Eq. 15
28:       $\tilde{u}_i^{(l)} \leftarrow \sum_{c=1}^C p_c \cdot a_{c,i}^{(l)}/n_c^{(l)}$ for all $i \in [M]$          // Eq. 12
29:       $\tilde{\phi}_i^{(l)} \leftarrow \tanh\big((\bar{K}/M)/(\tilde{u}_i^{(l)} + \epsilon) - 1\big)$          // Eq. 13
30:       $\phi_i^{(l,t+1)} \leftarrow (1-\zeta)\tilde{\phi}_i^{(l)} + \zeta\phi_i^{(l,t)}$          // Momentum update
31:    **end for**
32: **end for**
33: **Return:** $\Theta^{(T)}$

---

### D.2. Dynamic Modulated Routing (DMR)

Algorithm 2 details Dynamic Modulated Routing. Given token $x$, the router computes raw scores and selects top-$N_p$ experts as candidates $\mathcal{T}$. Modulation parameters $\phi$ are applied only within $\mathcal{T}$, preserving semantic relevance while rebalancing utilization. The final activation set $\mathcal{A}_c(x)$ contains the top-$K_c$ experts from modulated scores, with gating weights normalized via softmax.

---

**Algorithm 2** Dynamic Modulated Routing (DMR)

---

**Require:** Token $x \in \mathbb{R}^d$, router $W^{(r)} \in \mathbb{R}^{M \times d}$, modulation $\phi \in \mathbb{R}^M$, sparsity $K_c$, candidate size $N_p$
**Ensure:** Activation set $\mathcal{A}_c(x)$, gating weights $\gamma \in \mathbb{R}^M$

1: $\mathbf{s} \leftarrow W^{(r)}x$       // Raw router scores, $\mathbf{s} \in \mathbb{R}^M$
2: $\mathcal{T} \leftarrow \text{TOP-}N_p(\mathbf{s})$       // Candidate set: top-$N_p$ by semantic relevance
3: $m_i \leftarrow s_i + \phi_i \cdot \mathbf{1}[i \in \mathcal{T}]$ for all $i \in [M]$       // Modulate candidates only
4: $\mathcal{A}_c(x) \leftarrow \text{TOP-}K_c(\mathbf{m})$       // Select top-$K_c$ experts
5: $\gamma_i \leftarrow \frac{\exp(m_i)}{\sum_{j \in \mathcal{A}_c(x)} \exp(m_j)} \cdot \mathbf{1}[i \in \mathcal{A}_c(x)]$ for all $i$       // Sparse gating
6: **Return:** $\mathcal{A}_c(x), \gamma$

---

### D.3. Universal Pseudo-Gradient (PG)

Algorithm 3 describes client-side pseudo-gradient injection. Activated experts receive their actual gradients, while non-activated experts receive scaled pseudo-gradients from the server's buffer $\tilde{\mathbf{G}}^{(t)}$. The scaling factor $\rho_c = \sqrt{\bar{K}/K_c}$ amplifies signals for low-resource clients.

---

**Algorithm 3** Universal Pseudo-Gradient (PG) Injection

---

**Require:** Gradients $\{\nabla_{\Theta_i^{(e)}} \mathcal{L}_c\}_{i=1}^M$, activation set $\mathcal{A}_c$, pseudo-gradient buffer $\tilde{\mathbf{G}}^{(t)}$, scaling $\rho_c = \sqrt{\bar{K}/K_c}$
**Ensure:** Modified gradient $\hat{\mathbf{g}}_c = \{\hat{g}_{c,i}\}_{i=1}^M$

1: **for** $i = 1, \ldots, M$ **do**
2:    **if** $i \in \mathcal{A}_c$ **then**
3:       $\hat{g}_{c,i} \leftarrow \nabla_{\Theta_i^{(e)}} \mathcal{L}_c$       // Activated: use actual gradient
4:    **else**
5:       $\hat{g}_{c,i} \leftarrow \rho_c \cdot \tilde{G}_i^{(t)}$       // Non-activated: scaled pseudo-gradient
6:    **end if**
7: **end for**
8: **Return:** $\hat{\mathbf{g}}_c$

---

# E. Additional Experimental Details

### E.1. Benchmarks

We evaluate our method on 2 benchmarks: Commonsense-15K and TeleQuAD (in telecommunication domain).

**Commonsense-15K.** We fine-tune on the Commonsense-15K dataset (Hu et al., 2023), a curated set of 15,119 instruction-tuning samples. We evaluate on the test sets of 8 commonsense reasoning datasets: ARC-Challenge, ARC-Easy, BoolQ, HellaSwag, OpenBookQA, PIQA, Social IQa, and WinoGrande. Table 10 summarizes the evaluation benchmark statistics.

*Table 10.* Evaluation benchmark statistics for commonsense reasoning. Training uses Commonsense-15K (15,119 samples).

| Dataset | Test Size | Task Type |
|---|---|---|
| ARC-Challenge (Clark et al., 2018) | 1,172 | Multiple Choice |
| ARC-Easy (Clark et al., 2018) | 2,376 | Multiple Choice |
| BoolQ (Clark et al., 2019) | 3,270 | Yes/No QA |
| HellaSwag (Zellers et al., 2019) | 10,042 | Sentence Completion |
| OpenBookQA (Mihaylov et al., 2018) | 500 | Multiple Choice |
| PIQA (Bisk et al., 2020) | 1,838 | Physical Reasoning |
| Social IQa (Sap et al., 2019) | 1,954 | Social Reasoning |
| WinoGrande (Sakaguchi et al., 2021) | 1,267 | Coreference Resolution |

**TeleQuAD.** We additionally evaluate on TeleQuAD (Holm, 2021), a question-answering dataset in the telecommunications domain derived from 3GPP technical specifications. The dataset contains 4,262 answerable QA pairs across 536 documents, following a SQuAD-like extractive QA format. This domain-specific benchmark tests the method's effectiveness on specialized technical content and serves to evaluate robustness under both IID and non-IID data distributions. Table 11 summarizes the TeleQuAD dataset statistics used for domain-specific evaluation.

*Table 11.* TeleQuAD dataset statistics for telecommunications domain evaluation. The dataset follows SQuAD format with extractive question-answer pairs derived from 3GPP technical specifications.

| Statistic | Value |
|---|---|
| Domain | Telecommunications (3GPP) |
| Task Type | Extractive QA |
| Data Format | SQuAD-compatible |
| Source Documents | 3GPP Specifications |
| Training Samples | 1,018 question-answer pairs |
| Test Samples | 1,003 question-answer pairs. |
| Evaluation Metric | BERTScore F1 |

**Non-IID TeleQuAD partitioning.** For the non-IID TeleQuAD setting, we use document-level Dirichlet partitioning with concentration parameter $\alpha = 0.1$. Specifically, question-answer pairs are grouped by their source 3GPP document, and each client's mixture over document groups is sampled from a Dirichlet distribution. This creates strong topic/domain skew while preserving realistic document-level heterogeneity. Dirichlet partitioning is a standard protocol for controlled non-IID FL evaluation (Hsu et al., 2019; Yurochkin et al., 2019). Commonsense-15K is a curated instruction-tuning mixture without consistent class IDs across all tasks, so class-Dirichlet partitioning is not well-defined; we therefore report its multi-domain benchmark performance and use TeleQuAD for controlled IID/non-IID comparisons.

## E.2. Training Prompts

We use the Alpaca-style instruction format for fine-tuning. Table 12 summarizes the prompt templates for each task. The placeholder {instruction} is replaced with the task-specific question, and {context} (for TeleQuAD) contains the relevant paragraph from 3GPP telecommunications documentation.

*Table 12.* Prompt templates for instruction fine-tuning.

| Task | Prompt Template |
|---|---|
| Commonsense Reasoning | Below is an instruction that describes a task.  Write a response that appropriately completes the request.

### Instruction:
{instruction}

### Response: |
| TeleQuAD (Extractive QA) | Below is an instruction that describes a task, paired with context that provides further information.  Write a response that appropriately completes the request.

### Instruction:
{instruction}

### Context:
{context}

### Response: |

## E.3. Baseline Methods

We compare UB-SMoE against two categories of federated fine-tuning methods designed for system heterogeneity.

**Heterogeneous LoRA-rank Methods:**

- **HetLoRA** (Cho et al., 2024): introduces a heterogeneous rank allocation strategy combined with rank self-pruning. This method acts as a regularizer, enabling the aggregation of diverse client updates via a sparsity-weighted mechanism

to achieve superior convergence.

- **FlexLoRA** (Bai et al., 2024): aggregates heterogeneous LoRA adapters by constructing a full-size global weight matrix from weighted local contributions.

- **FLoRA** (Wang et al., 2024b): utilizes a stacking-based aggregation mechanism to seamlessly integrate heterogeneous LoRA adapters across clients. This method avoids aggregation noise by stacking local low-rank matrices ($A_k$ and $B_k$) to construct a precise global update, thereby enhancing convergence and model performance.

- **FLoRIST** (Ramesh & Dass, 2025): performs SVD directly on stacked local adapters. This method identifies and retains only the most informative components to construct compact global adapters.

**Heterogeneous Sparsity Methods:**

- **SMoE-LLB** (Zoph et al., 2022): addresses training instabilities in sparse models through a novel router z-loss and architectural heuristics. This approach also optimizes the trade-off between parameter sparsity and computational cost by utilizing top-2 routing and specific capacity factor adjustments.

- **A$^3$SMoE** (Tran et al., 2025): extends Sparse Mixture-of-Experts by allowing clients to activate a variable number of experts proportional to their computational budgets. This method employs an activation-aware aggregation strategy that weights global updates based on client-specific expert utilization frequencies.

**Homogeneous LoRA-rank Methods**

- **FedEx-LoRA** (Singhal et al., 2024): achieves exact aggregation in federated fine-tuning by correcting the global model with a residual error term derived from the deviation between ideal and actual LoRA updates.

- **FFA-LoRA** (Sun et al., 2024): addresses the discordance between joint local optimization and separate global aggregation in federated LoRA by freezing the randomly initialized matrix $A$ and only fine-tuning the zero-initialized matrix $B$.

- **FedIT** (Zhang et al., 2024): employs LoRA to freeze pre-trained weights and train only low-rank matrices on local devices, significantly reducing computational and communication costs.

- **FedSB** (Singhal et al., 2025): adapts LoRA-SB framework (Ponkshe et al., 2024) by training a small $r \times r$ matrix between frozen, optimally initialized low-rank adapters. This formulation enables exact aggregation via direct averaging of the central matrix, rendering communication costs independent of the client count while preserving high model expressivity.

All baseline methods use the same base model architecture and are trained for the same number of communication rounds to ensure fair comparison.

### E.4. System and Resource

We conduct all experiments on a high-performance computing (HPC) node equipped with two NVIDIA H100 96GB GPUs. Our implementation is based on PyTorch 2.4.0 with CUDA 12.4.1.

### E.5. Implementation Details

**Base Model.** We use OLMoE-1B-7B-0924 (Muennighoff et al., 2024) and OLMo-1B (Groeneveld et al., 2024) as the base models for all experiments. The base architecture consists of vocabulary size of $50,304$ tokens, hidden dimension $d = 2,048$, intermediate dimension $l = 2,048$, 16 transformer decoder layers, 16 attention heads with 16 key-value heads (standard multi-head attention, not grouped-query attention), maximum sequence length of 4,096 positions, RoPE positional embeddings with $\theta = 10,000$, SiLU activation function, RMSNorm with $\epsilon = 10^{-5}$, and no attention bias or dropout. The model weights remain frozen during federated fine-tuning, with only LoRA adapters being trained.

**SMoE Architecture.** The base model natively employs SMoE layers in place of standard feed-forward networks. The architecture contains $L = 16$ SMoE layers, each with $M = 64$ experts. The default number of activated per

*Table 13.* Hyperparameter configurations for heterogeneous federated fine-tuning experiments. For each setting and method, we specify the client-specific sparsity level $K_c$ (number of activated experts) and LoRA rank. UB-SMoE varies $K_c$ while fixing rank $r$; baseline methods either vary rank $r_c$ with fixed sparsity or operate on dense models. Configurations are calibrated to match trainable parameter counts across comparable budget levels.

| Settings | Applied Methods | Model | Resource Level | Detailed Configuration | # Trainable Params | FLOPs |
|---|---|---|---|---|---|---|
| | Ours, A$^3$SMoE, SMoE-LLB | MoE (OLMoE-1B-7B) | $\beta_1$ | TopK=1, LoRA_r=20 | $8.19E+06$ | $4.02E+11$ |
| | | | $\beta_2$ | TopK=2, LoRA_r=20 | $1.11E+07$ | $4.72E+11$ |
| | | | $\beta_3$ | TopK=4, LoRA_r=20 | $1.70E+07$ | $5.95E+11$ |
| | | | $\beta_4$ | TopK=8, LoRA_r=20 | $2.57E+07$ | $8.25E+11$ |
| System heterogeneity | FlexLoRA, HetLoRA, FLoRA, FLoRIST | MoE (OLMoE-1B-7B) | $\beta_1$ | TopK=8, LoRA_r=6 | $8.65E+06$ | $6.94E+11$ |
| | | | $\beta_2$ | TopK=8, LoRA_r=8 | $1.15E+07$ | $7.13E+11$ |
| | | | $\beta_3$ | TopK=8, LoRA_r=12 | $1.63E+07$ | $7.33E+11$ |
| | | | $\beta_4$ | TopK=8, LoRA_r=20 | $2.57E+07$ | $7.56E+11$ |
| | FlexLoRA, HetLoRA, FLoRA, FLoRIST | Dense (OLMo-1B) | $\beta_1$ | LoRA_r=12 | $9.04E+06$ | $8.31E+11$ |
| | | | $\beta_2$ | LoRA_r=16 | $1.21E+07$ | $8.49E+11$ |
| | | | $\beta_3$ | LoRA_r=24 | $1.81E+07$ | $8.70E+11$ |
| | | | $\beta_4$ | LoRA_r=40 | $3.01E+07$ | $8.93E+11$ |
| System homogeneity | FedEx, FFA-LoRA, FedIT, Fed-SB, Ours (UB-SMoE) | MoE (OLMoE-1B-7B) | $\beta_1$ | TopK=1, LoRA_r=20 | $8.19E+06$ | $4.02E+11$ |
| | | | $\beta_2$ | TopK=2, LoRA_r=20 | $1.11E+07$ | $4.72E+11$ |
| | | | $\beta_3$ | TopK=4, LoRA_r=20 | $1.70E+07$ | $5.95E+11$ |
| | | | $\beta_4$ | TopK=8, LoRA_r=20 | $2.57E+07$ | $8.25E+11$ |

token is $K_{\max} = 8$. For Dynamic Modulated Routing, the candidate set size is $N_p = 2$, representing the top experts considered for modulation. Each expert follows a gated feed-forward network architecture with `gate_proj`, `up_proj`, and `down_proj` components, using intermediate dimension $2 \times 2048$. The router is implemented as a linear projection from hidden dimension to number of experts ($2048 \rightarrow 64$).

**LoRA and Top-K Configuration.** We apply LoRA adaptation to both the attention layers and the expert networks. The target modules include `q_proj`, `k_proj`, `v_proj`, `o_proj` for attention, and `gate_proj`, `up_proj`, `down_proj`, `gate` for the SMoE components. We use standard LoRA with scaling parameter $\alpha = 20$ and dropout rate of $0.0$.

**Client Computation Budgets.** We assign heterogeneous computation budgets $\beta_c \in \{\beta_1, \beta_2, \beta_3, \beta_4\}$ corresponding to resource ratios $\{0.125, 0.25, 0.5, 1.0\}$. The distribution is uniform across all clients. Detailed Top-K and LoRA ranks configurations are provided in Tab. 13.

**Heterogeneous Sparsity (OLMoE-1B-7B).** For UB-SMoE and sparsity-based baselines, we fix $r = 20$ across all clients and vary the number of activated experts as $K_c \in \{1, 2, 4, 8\}$ for $\beta_1$ through $\beta_4$, yielding trainable activated parameters ranging from $8.19 \times 10^6$ ($K_c = 1$) to $2.57 \times 10^7$ ($K_c = 8$) as detailed in Tab. 13.

**Heterogeneous LoRA Ranks (OLMoE-1B-7B).** For rank-based baselines on the SMoE model, we fix $K = 8$ and vary client-specific ranks as $r_c \in \{6, 8, 12, 20\}$ for $\beta_1$ through $\beta_4$, so that the number of trainable activated parameters is approximately the same as the corresponding heterogeneous sparsity configurations.

**Heterogeneous LoRA Ranks (Dense OLMo-1B).** For fair comparison on the dense baseline, we set $r_c \in \{12, 16, 24, 40\}$ for $\beta_1$ through $\beta_4$, calibrated to match the trainable parameter counts of the corresponding SMoE configurations.

**Federated Learning Configuration.** The federated system consists of $C = 8$ clients communicating over $T = 10$ rounds by default, with extended experiments running up to $T = 20$ rounds. Each client performs $\Gamma = 0.4$ local epochs per round via gradient accumulation. Data is distributed across clients using either sequential or random splitting strategies. We employ a merged layers communication pattern where only modified layers are transmitted, comprising approximately 2–10% of the total model parameters. Aggregation weights follow the standard formulation $p_c = |\mathcal{D}_c| / \sum_{c'} |\mathcal{D}_{c'}|$.

**Optimization Hyperparameters.** We use the AdamW optimizer with learning rate $\eta = 2 \times 10^{-4}$, momentum parameters $(\beta_1, \beta_2) = (0.9, 0.95)$, epsilon $10^{-5}$, and weight decay $0.01$. The per-device batch size is $B = 8$ with 2 gradient accumulation steps, yielding an effective batch size of 16. We use a constant learning rate scheduler with no warmup or decay.

For Dynamic Modulated Routing (DMR), the learnable modulation parameters are constrained to the range $[\phi_{\min}, \phi_{\max}] = [-1.0, 1.0]$. The mechanism preserves the top-2 experts from the original router selection during modulation. The server

updates modulation vectors using the momentum factor $\zeta = 0.9$.

For Universal Pseudo-Gradient (PG), we apply gradient clipping with a threshold of 2.0. The K-aware scaling factor follows $\rho_c = \sqrt{8/K_c}$, providing stronger scaling for clients with smaller $k$ values.

**Reproducibility.** Random seeds are fixed at 42 for all experiments. We will release our code and pretrained checkpoints upon acceptance.

# F. Complexity Analysis

We provide a comprehensive analysis of the computational and communication complexity of UB-SMoE. We denote the number of experts per SMoE layer as $M$, the number of layers as $L$, and the LoRA rank as $r$. The model hidden dimension is $d$, while $l$ denotes the intermediate dimension of the feed-forward network within each expert. The number of clients is $C$, local iterations per round is $\Gamma$, and batch size is $\mathcal{B}$.

## F.1. Client-side Computation Complexity

The client-side computation consists of three components: Dynamic Modulated Routing (DMR), expert forward/backward passes, and pseudo-gradient injection.

**Dynamic Modulated Routing (DMR) Computation .** For each input $x \in \mathbb{R}^d$ at each SMoE layer, DMR (Algorithm 2) performs the following operations:

- Computing $s = W^{(r)}x$ requires $\mathcal{O}(M \cdot d)$ operations.

- Identifying the top-$N_p$ experts from $M$ scores requires $\mathcal{O}(M)$

- Applying modulation to $N_p$ candidates and computing softmax over $M$ experts requires $\mathcal{O}(M)$ operations.

- Selecting top-$K_c$ experts from modulated probabilities requires $\mathcal{O}(M + K_c)$

The total DMR overhead per input per layer is $\mathcal{O}(M \cdot d)$, which is dominated by the router score computation.

**Expert Computation.** For $K_c$ activated expert with LoRA adaptation, the forward pass requires $\mathcal{O}(K_c \cdot d \cdot l)$. The backward pass also requires $\mathcal{O}(K_c \cdot d \cdot l)$ for $K_c$ activated experts.

**Universal Pseudo-Gradient (PG) Injection.** During the backward pass, the pseudo-gradient mechanism (Algorithm 3) injects scaled gradients for non-activated experts. LoRA parameters consist of matrices $B \in \mathbb{R}^{d \times r}$ and $A \in \mathbb{R}^{r \times l}$, so the parameter count per expert is proportional to $r(d + l)$. For each of the $M - K_c$ non-activated experts, PG injection requires copying and scaling $(M - K_c)$ vectors, costing $\mathcal{O}((M - K_c) \cdot d \cdot r)$. The PG overhead per local iteration is $\mathcal{O}(L \cdot (M - K_c) \cdot r \cdot (d + l))$.

**Total Client-side Complexity.** The total computational cost per communication round for client $c$ is:

$$\mathcal{C}_{\text{client}} = \mathcal{O}\left(\Gamma \cdot \mathcal{B} \cdot L \cdot (Md + K_c \cdot d \cdot l) + \Gamma \cdot L \cdot M \cdot r \cdot (d + l)\right). \tag{57}$$

This computation complexity shows the advantage of UB-SMoE. The client-side computation scles with the client's computational budget $\beta_c$ through the sparsity level $K_c = \lfloor K_{\max} \cdot \beta_c \rfloor$. In standard dense fine-tuning (e.g., homogeneous LoRA), each client must compute the full feed-forward network for every token, incurring a cost of $\mathcal{O}(d \cdot l)$ per token per layer. In contrast, UB-SMoE clients only compute $K_c$ experts, reducing this to $\mathcal{O}(K_c \cdot d \cdot l)$. Table 14 illustrates the computation scaling across different client budget levels in UB-SMoE.

When compared with heterogeneous LoRA-rank methods (e.g., HetLoRA, FlexLoRA), our UB-SMoE also shows better effective client-side compution. Although these methods adapt to client resources by varying the rank $r_c$, the computation reduction is limited because the dominant cost lies in the dense FFN layers, not the LoRA adapters. As shown in Figure 3, the reduction of these methods is only $\sim 5\%$. In constrat, with less activated $K_c$ experts, low-resource clients in our UB-SMoE achieve substantial reduction ($\sim 45$ for $\beta_1$). However, UB-SMoE introduces additional overhead which arises from PG injection. This also explain why for high-resource clients, the computational cost is slightly higher.

*Table 14.* Illustration of client-side computation reduction in UB-SMoE.

| Budget Level | $\beta_c$ | $K_c$ | Expert FLOPs | Reduction |
|:---:|:---:|:---:|:---:|:---:|
| $\beta_4$ (High) | 1.0 | 8 | $8 \cdot d \cdot l$ | 0% |
| $\beta_3$ | 0.5 | 4 | $4 \cdot d \cdot l$ | 50% |
| $\beta_2$ | 0.25 | 2 | $2 \cdot d \cdot l$ | 75% |
| $\beta_1$ (Low) | 0.125 | 1 | $1 \cdot d \cdot l$ | 87.5% |

## F.2. Server-side Computation Complexity

The server performs aggregation and updates modulation parameters at each communication round.

**Aggregation.** The server aggregates LoRA parameters from $C$ clients via weighted averaging. The total number of trainable parameters is $|\Theta| = \mathcal{O}(L \cdot M \cdot r \cdot (d+l))$. The aggregation complexity is: $\mathcal{O} = (C \cdot L \cdot M \cdot r \cdot (d+l))$.

**PG Buffer Update.** Computing PG requires element-wise substraction and scaling; thus this costs $\mathcal{O}(L \cdot M \cdot r \cdot (d+l))$ operations.

**Parameter Update.** The server computes global utilization statistics and updates modulation vectors. For utilization aggregation, the server aggregates activation counts from $C$ clients for $M$ experts across $L$ layers, which costs $\mathcal{O}(C \cdot L \cdot M)$. To update modulation, the server computes $\tanh$ and momentum updates for $L \cdot M$ parameters, which requires $\mathcal{O}(L \cdot M)$ operations.

**Total Server-side Complexity.** The total server computation per round is:

$$\mathcal{C}_{\text{server}} = \mathcal{O}(C \cdot L \cdot M \cdot r \cdot (d+l) + L \cdot M). \tag{58}$$

## F.3. Communication Complexity

### F.3.1. CLIENT-TO-SERVER COMMUNICATION

At each communication round, each client $c$ transmits updated local parameters and utilization statistics to the server.

**Model Parameters $\Theta_c^{(t,\Gamma)}$.** Each clinet in UB-SMoE transmits updates not only for $K_c$ activated experts but also the PGs for non-activated experts (Algorithm 3). For each expert $i$, the LoRA adaptation of model parameters consists of matrices $B_i^{(e)} \in \mathbb{R}^{d \times r}$ and $A_i^{(e)} \in \mathbb{R}^{r \times l}$. The total parameter count for each client's updates is: $L \cdot M \cdot r \cdot (d+l)$.

**Utilization Statistics.** Clients must also transmit the activation count vector $a_c^{(l)} \in \mathbb{R}^M$ and token counts $n_c^{(l)}$ to the server. This requires $L \cdot (M+1)$ scalar values per client.

**Total client-to-server communication per round is:**

$$\mathcal{C}_{C \to S} = \mathcal{O}(L \cdot M \cdot r \cdot (d+l) + L \cdot (M+1)) \tag{59}$$

The client-to-server communication is *independent* of the client's computational budget $\beta_c$ and sparsity level $K_c$. To ensure the exper utilization, UB-SMoE introduces additional communication costs per client ($L \cdot (M+1)$).

### F.3.2. SERVER-TO-CLIENT COMMUNICATION

The server broadcasts the global model parameters, PG buffer, and modulation parameters.

**Global Model Parameters ($\Theta^{(t)}$).** The server broadcasts the aggregated LoRA parameters contains router and all $M$ experts. This requires $L \cdot M \cdot r \cdot (d+l)$ parameters.

**PG Buffer ($\tilde{G}^{(t)}$).** To enable the PG injection locally, the server must broadcast the global pseudo-gradient buffer. Each buffet contain gradients for $M$ experts, with each expert gradient having dimensions matching the LoRA parameters: $r \cdot (d+l)$. The total PF buffer size is: $L \cdot M \cdot r \cdot (d+l)$.

**Modulation Parameters ($\phi^{(l)}$).** The server transmits the updated modulation vectors, which requires $L \cdot M$ scalar values.

**Total server-to-client communication per round is:**

$$\mathcal{C}_{S \to C} = \mathcal{O}(2 \cdot L \cdot M \cdot r \cdot (d + l) + L \cdot M). \tag{60}$$

Regarding the server-to-client communication, UB-SMoE introduces additional communication overhead due to the PG buffer $(L \cdot M \cdot r \cdot (d + l))$ and modulation parameters $(L \cdot M)$.

F.3.3. NUMERICAL COMMUNICATION-COMPUTATION TRADE-OFF

Table 15 complements the asymptotic communication analysis with numerical upload and download costs. UB-SMoE has the same upload as $A^3$SMoE but a larger download due to the pseudo-gradient buffer. This is the main communication trade-off of UB-SMoE. However, for the low-resource budget $\beta_1$, UB-SMoE reduces FLOPs by 42% relative to LoRA-rank baselines and improves average accuracy by $8.6\times$ over the strongest LoRA-rank baseline, FlexLoRA. FLOPs are a hardware-agnostic proxy for compute budget commonly used in scaling and resource-constrained learning analyses (Pfeiffer et al., 2023; Kaplan et al., 2020); actual latency and energy depend on the device, kernel implementation, memory bandwidth, and communication stack.

*Table 15.* Numerical communication, computation, and low-resource accuracy trade-off. FLOPs and accuracy correspond to the lowest budget $\beta_1$ on Commonsense-15K.

| Method | Avg. upload | Avg. download | FLOPs ($\beta_1$) | Accuracy ($\beta_1$) |
|---|---|---|---|---|
| UB-SMoE | 370 MB | 730 MB | $4.02 \times 10^{11}$ | 0.3936 |
| $A^3$SMoE | 370 MB | 370 MB | $4.02 \times 10^{11}$ | 0.3629 |
| FLoRA | 212.75 MB | 212.75 MB | $6.94 \times 10^{11}$ | 0.0094 |
| HetLoRA | 212.75 MB | 212.75 MB | $6.94 \times 10^{11}$ | 0.0079 |
| FlexLoRA | 212.75 MB | 212.75 MB | $6.94 \times 10^{11}$ | 0.0456 |
| FLoRIST | 212.75 MB | 370 MB | $6.94 \times 10^{11}$ | 0.0112 |

## F.4. Comparison with other Heterogeneous Methods

F.4.1. COMPARISON WITH HETEROGENEOUS LORA-RANK METHODS

Heterogeneous LoRA-rank methods (FloRA (Wang et al., 2024b), HetLoRA (Cho et al., 2024), FlexLoRA (Bai et al., 2024), and FLoRIST (Ramesh & Dass, 2025)) address system heterogeneity by assigning client-specific LoRA ranks $r_c$ based on computational capabilities.

**Client-side Computation.** The client-specific computation scales with the rank value $r_c$. For most heterogenerous LoRA-rank methods (FloRA, FlexLoRA, and FLoRIST), the client-side computation is:

$$\mathcal{C}_{\text{client}}^{\text{LoRA}} = \mathcal{O}(\Gamma \cdot \mathcal{B} \cdot L \cdot (d \cdot l + r_c(d + l)). \tag{61}$$

The term $d \cdot l$ corresponds to the dense feed-forward network (FFN) computation, which remains unchanged regardless of the client's rank $r_c$. The term $r_c(d + l)$ is the LoRA adaptation specific to $r_c$ of each client. Since $r_c \ll \min(d, l)$, the LoRA-specific computation is negligible compared to FFN cost. This explains why these methods achieve only $\sim 5\%$ computation reduction even at the lowest budget $B_1$ (Figure 3). Regarding HetLoRA, the computation complexity for each client is: $\mathcal{C}_{\text{client}}^{\text{HetLoRA}} = \mathcal{O}(\Gamma \cdot \mathcal{B} \cdot L \cdot (d \cdot l + r_c(d + l)) + L \cdot r_c(d + l))$. The rank self-pruning mechanism introduces additional overhead, which requires computing singular value decomposition or gradient-based importance scores for the LoRA matrices.

In contrast, our UB-SMoE alters the computation structure, in which the expensive FFN computation now scales with $K_c$. For a low-resource client (e.g., $K_c = 1$), this represents a significant reduction in the computational cost.

**Server-side Computation.** To aggregate different rank size, heterogeneous LoRA methods propsoe various aggregation strategies, which lead tosignicant variation in server-side complexity.

- **FLoRA** employs a stacking-based aggregation that directly stacks the heterogeneous LoRA matrices from all clients. This requires simple weighted averaging with complexity $\mathcal{C}_{\text{server}}^{\text{FLoRA}} = \mathcal{O}(C \cdot L \cdot r_{\max}(d + l))$.

- **HetLoRA** performs sparsity-weighted aggregation that accounts for the pruned dimensions from each client. The server must reconstruct the full delta model for each client to compute its singular values, requiring $\mathcal{C}_{\text{server}}^{\text{HetLoRA}} = \mathcal{O}(C \cdot L \cdot r_{\max} \cdot d \cdot l)$.

- **FlexLoRA** needs to reconstruct full-size matrices and performs SVD to redistribute this aggregated update back to client-specific ranks. The SVD decomposition of the $d \cdot l$ matrix dominates the cost at $\mathcal{C}_{\text{server}}^{\text{FlexLoRA}} = \mathcal{O}(C \cdot L(d^2 \cdot l + r_{\max} \cdot d \cdot l))$, which scales quadratically with dimension $d$.

- **FLoRIST** performs SVD directly on the stacked adapter matrices rather than the full weight matrix. Given stacked matrices with total rank $R = \sum_c r_c$, the server computes a truncated SVD to extract the top-$p$ singular components. This requires $\mathcal{C}_{\text{server}}^{\text{FLoRIST}} = \mathcal{O}(C \cdot L \cdot R^2(d + l + R) + K \cdot p \cdot R(d + l)))$.

UB-SMoE requires $\mathcal{O}(C \cdot L \cdot M \cdot r(d + l) + L \cdot M)$ for aggregation and modulation updates. While this exceeds FLoRA's simple averaging, it avoids the expensive SVD operations required by FlexLoRA and FLoRIST.

**Communication Complexity.** Heterogeneous LoRA methods achieve communication efficiency through rank reduction at $\mathcal{O}(L \cdot r_c(d + l))$ per client. UB-SMoE incurs higher communication costs: $\mathcal{C}_{C \to S}^{\text{UB-SMoE}} = \mathcal{O}(L \cdot M \cdot r(d + l))$ and $\mathcal{C}_{S \to C}^{\text{UB-SMoE}} = \mathcal{O}(2 \cdot L \cdot M \cdot r(d + l))$.

A critical limitation of heterogeneous LoRA methods is that the merged weights $W_0 + \frac{\alpha}{r} BA$ *remain dense during inference*, leaving deployment latency unchanged. UB-SMoE preserves sparsity at inference time, enabling true resource-adaptive deployment.

### F.4.2. Comparison with Heterogeneous Sparsity Methods

**Computation Complexity.** A$^3$SMoE and UB-SMoE share the same clide-side complexity for forward and backward passes: $\mathcal{O}\big(\Gamma \cdot \mathcal{B} \cdot L(M \cdot d + K_c \cdot d \cdot l)\big)$. UB-SMoE introduces additional overhead from the pseudo-gradient mechanism at $\mathcal{O}(\Gamma \cdot L \cdot M \cdot r(d + l))$. Note that $B \cdot K_c \cdot d \cdot l \gg M \cdot r(d + l)$. For server-side computation, both methods require $\mathcal{O}\big(\Gamma \cdot \mathcal{B} \cdot L(d \cdot l + r_c(d + l))\big)$ for parameter aggregation. UB-SMoE adds $\mathcal{O}(L \cdot M) \cdot r(d + l))$ for pseudo-gradient buffer computation and $\mathcal{O}(L \cdot M)$ for modulation updates.

**Communication Complexity.** Both methods have similar upload complexity. UB-SMoE requires $2\times$ the download bandwidth due to transmitting the pseudo-gradient buffer alongside model parameters, enabling continuous learning signals for all experts.

### F.4.3. Summary

Table 1 provides a comparison across all methods. The key insights from this complexity analysis are: (i) SMoE-based methods provide significant computaional reduction through conditional computation, unlike LoRA-rank methods where the dense FFN dominates; (ii) UB-SMoE trades increased communication for dramatically reduced computation at low-resource clients; and (iii) UB-SMoE maintains sparsity at inference, enabling deployment on resource-constrained devices.

Note that in this analysis, heterogeneous sparsity methods operate on SMoE architectures, while heterogeneous LoRA-rank methods are analyzed on dense transformer models. If LoRA-rank methods were applied to SMoE architectures, an additional factor of $M$ (the number of experts) would be introduced in all LoRA-related complexity terms, as each expert requires separate LoRA adaptation. This would further *increase the complexity* of LoRA-rank methods.

## G. Additional Experiments

### G.1. Analysis of Computational Cost (FLOPs))

This analysis empirically validates our central argument that heterogeneous LoRA-rank methods fail to provide meaningful computational savings for resource-constrained clients. We measure the total FLOPs required per forward-backward pass across four client capability levels ($\beta_1 - \beta_4$), using a fixed sequence length of 256 tokens. Detailed FLOPs measurements for each configuration are provided in Tab. 13.

As shown in Fig. 3, heterogeneous LoRA-rank methods achieve only a marginal reduction: even at the lowest budget $\beta_1$, computation decreases only $\sim 5\%$ (from $3.44 \times 10^{11}$ to $3.26 \times 10^{11}$ FLOPs). This confirms our analysis in Sec. F: LoRA-rank methods provide minimal computational savings since LoRA adaptation cost $\mathcal{O}(r_c(d + l))$ is negligible

compared to dense FFN computation $\mathcal{O}(d \cdot l)$. In contrast, heterogeneous sparsity methods achieve substantial reduction scaling with client budget: 69.4% at $\beta_3$, 53.5% at $\beta_2$, and 45.0% at $\beta_1$. This reduction directly stems from activating fewer experts, where computational cost scales as $\mathcal{O}(K_c \cdot d \cdot l)$.

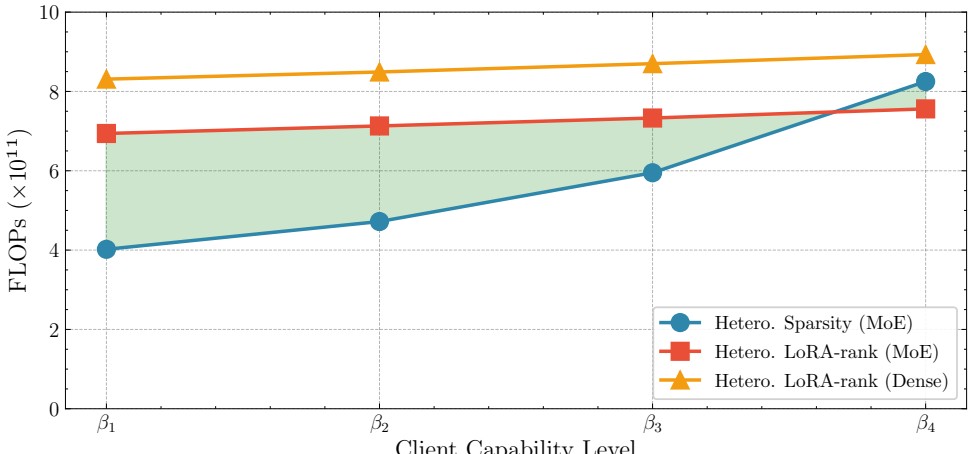

*Figure 3.* Computational requirements (in FLOPS) for two strategies aimed at adapting FMs to clients with varying resource levels, from high-capability ($\beta_4$) to low-capability ($\beta_1$). The heterogeneous LoRA-rank method reduces the rank of LoRA matrices. The Heterogeneous Sparsity method reduces the number of activated model components (experts).

These results demonstrate that sparsity methods are fundamentally better suited for resource-adaptive federated fine-tuning.

### G.2. Scalability Analysis with 32 Clients

To evaluate scalability beyond the standard 16-client setting, we conduct experiments with 32 clients under three system heterogeneity patterns. All experiments use Commonsense-15K training data and evaluate on 8 reasoning datasets.

#### G.2.1. UNIFORM DISTRIBUTION WITH PARTIAL PARTICIPATION

Table 16 presents results with 32 clients using uniform resource distribution and 50% participation rate per round. This setting tests the method's robustness when only half the clients participate in each federated round.

*Table 16.* 32-client results with uniform distribution and 50% participation rate. Performance averaged over all computation budgets.

| Dataset | HetLoRA | FlexLoRA | A³SMoE | SMoE-LLB | Ours |
|---|---|---|---|---|---|
| ARC-Challenge | 0.0960 | 0.1817 | **0.3174** | 0.3165 | 0.3129 |
| ARC-Easy | 0.1105 | 0.2127 | **0.4145** | 0.4052 | 0.4071 |
| BoolQ | 0.3448 | 0.3419 | 0.5785 | 0.5639 | **0.5814** |
| HellaSwag | 0.0635 | 0.0854 | 0.2245 | 0.2277 | **0.2355** |
| OpenBookQA | 0.1295 | 0.1655 | **0.3165** | 0.2925 | 0.3115 |
| PIQA | 0.2364 | 0.2493 | 0.4905 | 0.4853 | **0.5122** |
| Social IQa | 0.0861 | 0.1827 | **0.3912** | 0.3826 | 0.3888 |
| WinoGrande | 0.1375 | 0.2364 | 0.4649 | 0.4684 | **0.4795** |
| Average | 0.1505 | 0.2070 | 0.3998 | 0.3928 | **0.4036** |

#### G.2.2. LONG-TAIL HETEROGENEITY PATTERN

Table 17 shows results under a long-tail distribution where 75% of clients have minimal resources ($\beta_1$), 12.5% have medium-low ($\beta_2$), 6.25% have medium-high ($\beta_3$), and 6.25% have high resources ($\beta_4$). This pattern represents realistic scenarios where most edge devices are resource-constrained.

*Table 17.* 32-client results under long-tail heterogeneity (75% low-resource clients).

| Budget | HetLoRA | FlexLoRA | A³SMoE | SMoE-LLB | Ours |
|---|---|---|---|---|---|
| $\beta_1$ | 0.0087 | 0.0239 | 0.3554 | **0.3567** | 0.2190 |
| $\beta_2$ | 0.0505 | 0.0619 | **0.2007** | 0.1948 | 0.1854 |
| $\beta_3$ | 0.2807 | 0.4359 | 0.3010 | 0.2928 | **0.3715** |
| $\beta_4$ | 0.3928 | 0.5121 | 0.3381 | 0.3399 | **0.4430** |
| Average | 0.1832 | 0.2585 | 0.2988 | 0.2961 | **0.3047** |

#### G.2.3. REVERSE LONG-TAIL HETEROGENEITY PATTERN

Table 18 presents results under reverse long-tail distribution where 75% of clients have high resources ($\beta_4$), 12.5% have medium-high ($\beta_3$), 6.25% have medium-low ($\beta_2$), and 6.25% have low resources ($\beta_1$). This pattern

*Table 18.* 32-client results under reverse long-tail heterogeneity (75% high-resource clients).

| Budget | HetLoRA | FlexLoRA | A³SMoE | SMoE-LLB | Ours |
|---|---|---|---|---|---|
| $\beta_1$ | 0.0197 | 0.0243 | 0.2979 | 0.3340 | **0.3423** |
| $\beta_2$ | 0.0771 | 0.1234 | 0.3550 | 0.2825 | **0.3624** |
| 32 $\beta_3$ | 0.3174 | 0.4028 | 0.4684 | 0.4603 | **0.4721** |
| $\beta_4$ | 0.4826 | 0.4899 | 0.5380 | 0.4943 | **0.5319** |
| Average | 0.2242 | 0.2601 | 0.4148 | 0.3928 | **0.4272** |

tests performance when most clients can utilize full model capacity.

**Key Findings.**

- UB-SMoE achieves the best average performance across all three heterogeneity patterns.

- Under the challenging long-tail pattern, UB-SMoE outperforms the strongest baseline ($A^3$SMoE) by 2.0% despite 75% of clients being severely resource-constrained.

- Under reverse long-tail, UB-SMoE achieves 0.4272 average accuracy, outperforming $A^3$SMoE by 3.0%.

- With 50% participation rate, UB-SMoE maintains strong performance (0.4036), demonstrating robustness to partial client availability.

### G.3. How to determine the optimal pseudo-gradient scaling factor $\rho_c$?

The pseudo-gradient scaling factor $\rho_c$ is a critical component of our Universal Pseudo-Gradient (PG) mechanism, designed to compensate for the limited gradient coverage experienced by resource-constrained clients. This section systematically evaluates different scaling functions to determine the optimal formulation.

*Table 19.* Ablation study on techniques of the scaling factor $\rho$. The results are averages across 8 commonsense reasoning datasets.

| None | root square | linear | inverse root square | log |
|------|-------------|--------|---------------------|-----|
| 0.2953 | **0.4267** | 0.3270 | 0.3181 | 0.3549 |

Let $\bar{K} = \sum_c p_c K_c$ denote the weighted average system sparsity across all clients. We investigate five scaling strategies:

1. **None** (No scaling): $\rho_c = 1$

2. **Root Square** (Proposed): $\rho_c = \sqrt{\frac{\bar{K}}{K_c}}$

3. **Linear**: $\rho_c = \frac{\bar{K}}{K_c}$

4. **Inverse Root Square**: $\rho_c = \sqrt{\frac{K_c}{\bar{K}}}$

5. **Logarithmic**: $\rho_c = \log\left(\frac{\bar{K}}{K_c} + 1\right)$

Tab. 19, we can observe that: (i) Without any scaling, the system achieves the lowest performance (0.2953), confirming that scaling is essential for heterogeneous federated SMoE; (ii) The root square scaling significantly outperforms all alternatives with 0.4267 average accuracy. These results confirm that the root square scaling prevents excessive noise injection while still providing meaningful gradient signals to low-resource clients.

### G.4. What is the optimal $\phi$ range?

The parameter $\phi$ in our DMR mechanism directly influences expert selection by adjusting router logits. We investigate how this range affects the balance between two competing objectives: (i) correcting utilization imbalance by promoting under-utilized experts, and (ii) preserving semantic relevance by not forcing the selection of irrelevant experts. Tab. 20 presents the results across four range configurations. The optimal average performance (0.4267) is achieved with a range of $[-1.0, 1.0]$. This range provides the necessary flexibility to both promote under-utilized experts and penalize over-utilized ones without destabilizing the heterogeneous federated environment.

*Table 20.* Ablation study on the impact of the modulation parameter range $[\phi_{\min}, \phi_{\max}]$. Performance is averaged across all 8 datasets.

| $\phi \in [-0.5, 0.5]$ | $\phi \in [-1.0, 1.0]$ | $\phi \in [-2.0, 2.0]$ | $\phi \in [-0.5, 1.5]$ |
|------------------------|------------------------|------------------------|------------------------|
| 0.1114 | **0.4267** | 0.1303 | 0.2937 |

### G.5. Optimal number of preserved experts

In our DMR mechanism, we introduce a design choice: the number of experts to preserve from the original router's top selections before applying utilization-based modulation. This hyperparameter controls the trade-off between maintaining semantic relevance (based on the input token) and promoting utilization balance (based on client-specific computational budget). As shown in Tab.

*Table 21.* Ablation on the number of preserved experts

| 0 | 1 | 2 | 4 |
|---|---|---|---|
| 0.2684 | 0.2449 | **0.4267** | 0.2905 |

21, preserving 2 experts yields optimal performance (0.4267). No preservation (0.2684) risks selecting irrelevant experts, while excessive preservation (4 experts, 0.2905) limits DMR's ability to promote under-utilized experts.

### G.6. Analysis of Expert Utilization Balance Using Gini Coefficients

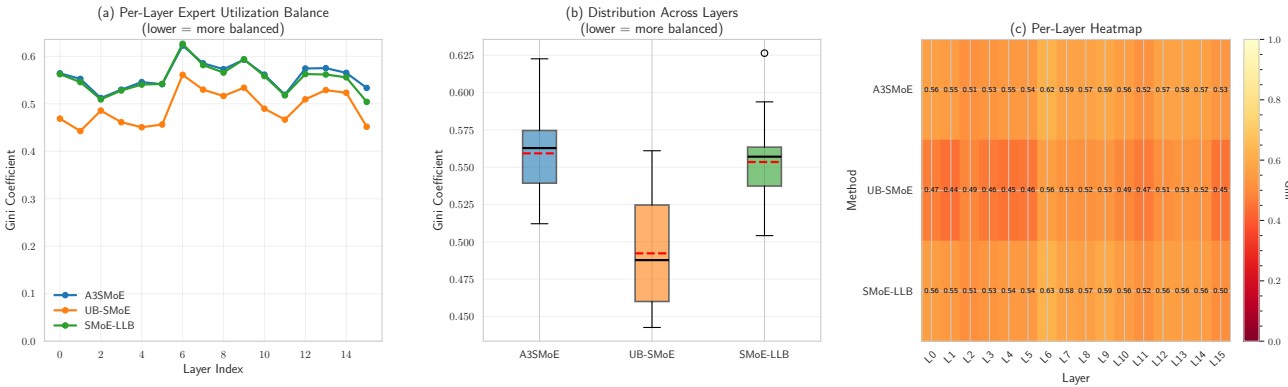

*Figure 4.* Expert Utilization Gini Analysis.

To complement our entropy-based analysis in Sec. 6.3.3, we employ the Gini coefficient as an alternative metric for evaluating expert utilization balance. Given an SMoE layer with $M$ experts and global utilization rates $\{\tilde{u}_i\}_i^M$, the Gini coefficient is computed as:

$$G = \frac{\sum_{i=1}^{M}\sum_{j=1}^{M}|\tilde{u}_i - \tilde{u}_j|}{2M\sum_{i=1}^{M}\tilde{u}_i} \tag{62}$$

$G$ is bounded in $[0, 1]$, where 0 indicates perfect equality (all experts equally utilized) and 1 represents maximal inequality (a single expert dominates all activations). From Fig. 4, several key observations can be made:

- UB-SMoE achieves consistently lower Gini coefficients across all layers: The average Gini coefficient for UB-SMoE is 0.49, representing a $12.5\%$ relative improvement over $A^3$SMoE (0.56) and $13.2\%$ over SMoE-LLB (0.56).

- UB-SMoE exhibits lower variance in balance across layers: Fig. 4(b) reveals that UB-SMoE not only achieves a lower median Gini coefficient but also demonstrates reduced variance across the 16 SMoE layers. The heatmap in Fig. 4(c) provides granular visibility into per-layer performance. This analysis indicates that our DMR mechanism provides consistent load balancing benefits.

- Middle layers exhibit higher utilization imbalance: Across all methods, the Gini coefficients are high across layers L6-L9. This is because early layers capture low-level patterns that are relatively uniform across inputs, while late layers perform task-specific processing. However, middle layers occupy a critical transition zone where surface-level features transform into abstract semantic representations. This abstraction process promotes expert specialization since routers learn to associate specific experts with semantic categories.

### G.7. Per-Layer Entropy Analysis Using Entropy

While Sec. 6.3.3 presents aggregate entropy statistics, understanding layer-wise behavior is essential for validating that DMR provides consistent balancing across the model depth.

Fig.5 presents per-layer expert utilization entropy across all 16 SMoE layers. UB-SMoE achieves consistently higher entropy across all layers, with an average of 3.72 compared to 3.52 for A³SMoE and 3.54 for SMoE-LLB, representing a 5.7%

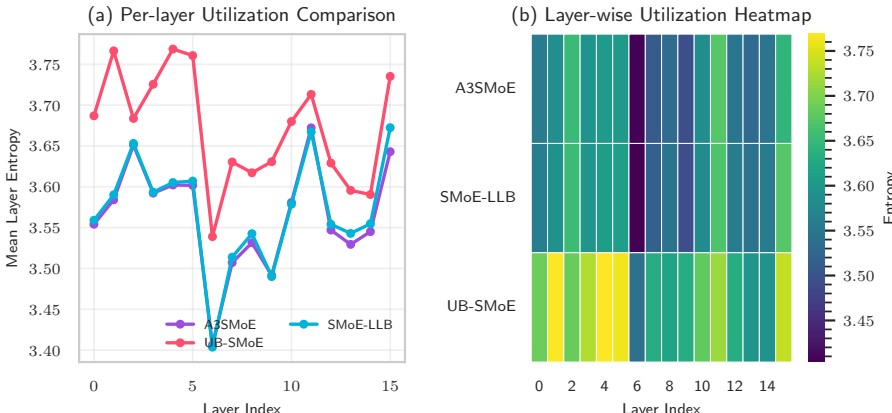

*Figure 5.* Layer-wise expert utilization entropy comparison. Higher entropy indicates more balanced utilization. UB-SMoE maintains superior entropy across all 16 SMoE layers.

and 5.1% relative improvement, respectively. The improvement is most pronounced in middle layers (L6–L10), where utilization imbalance tends to be most severe due to semantic abstraction inducing expert specialization. As discussed in Section G.G.6, these layers occupy a critical transition zone where semantic abstraction promotes expert specialization. The DMR mechanism effectively counteracts this tendency by promoting under-utilized experts within semantically relevant candidate sets. Furthermore, the heatmap reveals that UB-SMoE maintains lower variance in entropy across layers (standard deviation of 0.08) compared to A³SMoE (0.12) and SMoE-LLB (0.11).

### G.8. Analysis of $\phi$-utilization correlation

To validate the effectiveness of the $\phi$ values in modulating expert utilization, we analyze the correlation between the learned $\phi$ values and global expert utilization rates across training rounds. Particularly, For each expert $i$ at layer $l$, we record the global utilization rate $\tilde{u}_i^{(l)}$ (computed via Eq. 12) and the corresponding learned modulation parameter $\phi_i^{(l)}$.

We employ two correlation coefficients to characterize the $\phi$-utilization relationship. The Pearson correlation coefficient measures linear relationships:

$$r = \frac{\sum_{i=1}^{n}(\phi_i - \bar{\phi})(\tilde{u}_i - \bar{\tilde{u}})}{\sqrt{\sum_{i=1}^{n}(\phi_i - \bar{\phi})^2}\sqrt{\sum_{i=1}^{n}(\tilde{u}_i - \bar{\tilde{u}})^2}}, \quad (63)$$

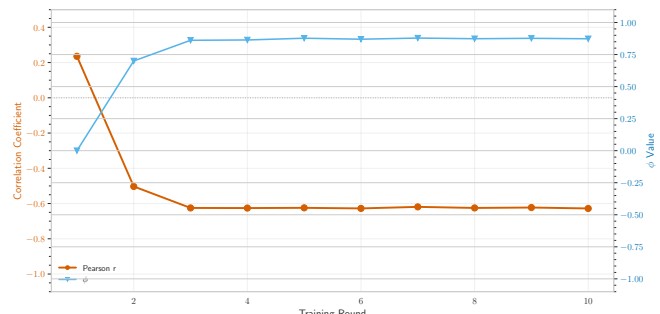

*Figure 6.* Dynamics of Dynamic Modulated Routing (DMR) across training rounds. We track the correlation between learned modulation parameters $\phi$ and global expert utilization rates, alongside the Pearson $r$ and Spearman $\rho$ correlation coefficients.

where $n = L \cdot M$ is the total number of expert-layer pairs, and $\bar{\phi}$ and $\bar{\tilde{u}}$ denote sample means. This coefficient ranges from $-1$ (perfect negative correlation) to $+1$ (perfect positive correlation).

From Fig. 6 and Fig. 7, we observe:

- (i) The correlation coefficients stabilize at approximately $r \approx -0.62$, indicating that DMR has converged to a consistent balancing policy. The moderate-to-strong negative correlation confirms that over-utilized experts receive negative modulation (reducing their selection probability) while under-utilized experts receive positive modulation (increasing their selection probability).

- (ii) The regression slope becomes increasingly negative over rounds, stabilizing around $-1.5$. This indicates that DMR applies proportionally stronger corrections as training progresses.

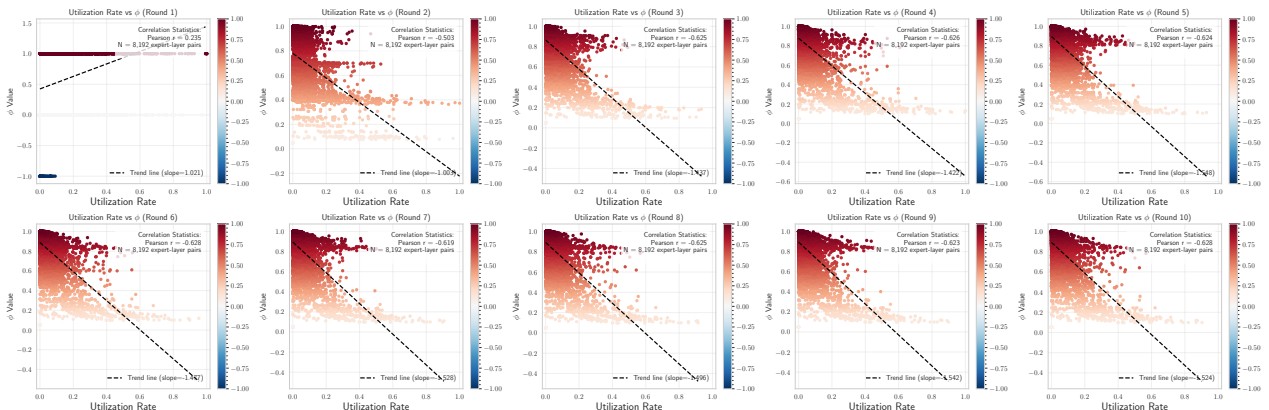

*Figure 7.* Correlation analysis between expert utilization rates and DMR modulation parameters $\phi$ over training. Each scatter plot visualizes 8,192 expert-layer pairs with corresponding Pearson ($r$) correlation coefficients. Trend lines (dashed) indicate the linear regression slope.

## H. Discussion and Limitations

Our work reveals that the fundamental bottleneck in heterogeneous federated fine-tuning lies not in adapter rank but in the dense feed-forward computation. While heterogeneous LoRA-rank methods achieve only $\sim$5% computation reduction even at the lowest budget level, UB-SMoE provides up to 45.0% reduction on low-resource clients by directly addressing this bottleneck through conditional computation. Furthermore, our work indicates that local load-balancing methods in standard SMoE are insufficient for heterogeneous federated fine-tuning. Therefore, the global utilization imbalance should be focused on to completely address the expert utilization imbalance problem.

Our method has two primary limitations. First, the effectiveness of DMR diminishes as the number of preserved experts approaches the client's activation budget $K_c$. For clients where $K_c \leq N_p$, the router entirely determines expert selection based on input token semantics, and the utilization-aware modulation provides no additional benefit. Moreover, the focus of this work is on addressing expert utilization imbalance and gradient sparsity in heterogeneous federated SMoE. While UB-SMoE requires $2\times$ download bandwidth for pseudo-gradient buffers (see Tab. 1), exploring communication-efficient variants (e.g., compression, quantization, selective transmission) is deferred to future work.

