# OpenReview forum: "UB-SMoE: Universally Balanced Sparse Mixture-of-Experts for Resource-adaptive Federated Fine-tuning of Foundation Models"
_ICML.cc/2026/Conference — ICML 2026 regular_

### Official Review · Reviewer_fErL · 2026-03-13

**Soundness:** 3
**Presentation:** 2
**Significance:** 2
**Originality:** 2
**Overall Recommendation:** 3
**Confidence:** 3

**Summary:**

This paper investigates federated fine-tuning of foundation models under heterogeneous client computational budgets, where existing heterogeneous LoRA-rank methods have certain limitations. To this end, the paper replaces rank-based adaptation methods with a Sparse Mixture-of-Experts (SMoE) architecture. However, when applied to heterogeneous federated fine-tuning, this architecture faces two key issues: first, imbalance in expert utilization; second, the non-differentiability of Top-K routing. The authors verify the existence of these issues through convergence analysis and show that their impact is more severe for low-resource clients. To address these problems, the paper proposes the UB-SMoE method, introducing Dynamic Modulated Routing (DMR) to resolve expert utilization imbalance, and Universal Pseudo-Gradient (PG) to reconstruct learning signals for non-activated experts, thereby ensuring that experts remain effective and balanced across heterogeneous clients. Finally, the experimental results validate the superior performance of the proposed method.

**Compliance With Llm Reviewing Policy:**

Affirmed.

**Final Justification:**

As I mentioned before, I still have concerns regarding the theoretical analysis, so I have decided not to change my score.

**Key Questions For Authors:**

1.	The theoretical analysis appears to target naive heterogeneous federated SMoE. For UB-SMoE, which incorporates DMR and PG, could the authors clarify which parts of the theory directly apply to the proposed algorithm and which parts mainly serve as motivation?
2.	In the paper, the update signal for non-activated experts is defined as a pseudo-gradient constructed from cross-round global parameter changes. Could the authors further explain why, under strong heterogeneity and non-IID conditions, this signal can still be regarded as reasonably aligned with the missing local gradients?
3.	Could the authors provide more evidence regarding robustness in the experimental section?

**Limitations:**

yes

**Strengths And Weaknesses:**

Strengths

1.	This paper focuses on resource-adaptive federated fine-tuning of foundation models, rather than being limited to parameter adaptation alone, and this research objective is of strong practical significance. The paper presents relatively thorough theoretical analysis and experimental validation around this problem, with a progressively developed argument and a method design that also demonstrates strong overall coherence.
2.	The overall logic of the paper is clear, and the problems to be addressed are explicitly stated. For the heterogeneous SMoE setting, the authors provide well-structured formal definitions, characterize with relative precision the two optimization mismatch issues, and further present corresponding convergence analysis results, proving both the optimization error terms and the lower bound of the error caused by bias.

Weaknesses

1.	The main theorem in the theoretical section is primarily established for naive federated SMoE, with the conclusion that it converges to a neighborhood under biased gradients, rather than directly providing a rigorous guarantee for the update mechanism of UB-SMoE. In addition, the theoretical analysis relies on several relatively strong assumptions. Although these assumptions are understandable in the context of theoretical derivation, they also mean that the practical support provided by the theorem should not be overstated. Moreover, most of the component techniques adopted in this paper are built on existing studies.
2.	The experimental section would be more convincing if it included more robustness analysis. For example, it would be helpful to further examine sensitivity to client sampling, performance under stronger non-IID settings, actual communication time cost, and results on other MoE backbone networks. In addition, if some key design details were presented in the main text rather than in the appendix, the overall argument would be more complete.
3.	The experimental results overall demonstrate certain advantages, but they may not be as stable as stated in the paper. UB-SMoE does not achieve the best performance on all tasks; therefore, a more accurate statement would be that it delivers better average performance, rather than establishing universal superiority. Similarly, the scalability results in Table 5 are more indicative of certain improvements, and the strength of the conclusion could also be expressed more cautiously.

---

> ### Author Rebuttal · Authors · 2026-03-30
>
> We thank Reviewer fErL for their constructive feedback and address each point below.
>
> **Weakness (W) 1: "Most component techniques are built on existing studies" and strong assumption**
>
> We appreciate the opportunity to clarify the novelty of our components relative to prior work. As discussed in Sec. 7, our DMR and PG are distinct from all existing methods: (i) Loss-Free Balancing (Wang et al., 2024a) uses local batch statistics, unsuitable for heterogeneous FL; our DMR uses global utilization statistics aggregated across clients and applies modulation only within a semantically pre-filtered candidate set. (ii) Default MoE (Panda et al., 2025) requires all experts within a single forward pass; PG provides learning signals via cross-round information for experts a client never activates — a capability no existing method offers. (iii) The synergistic DMR-PG cycle is itself a novel contribution with no precedent in federated or centralized SMoE literature. Moreover, our experiments have compared with all baselines, including LoRA-based methods, A³SMoE, and local-load balancing in default SMoE.
>
> Regarding assumptions: the PL condition adopted in our analysis is standard in FL convergence theory, used in recent works including FOCUS [1], EFSkip [2], and Li et al. [3]. Moreover, we fine-tune only lightweight LoRA adapters while keeping the pre-trained model frozen, meaning optimization occurs in a low-dimensional subspace where the PL condition is well-justified [4]. We will add this discussion to the revision.
>
> [1] arXiv:2503.20117 (NeurIPS-2025).
>
> [2] 10.1609/aaai.v39i15.33700 (AAAI-2025).
>
> [3] 10.1016/j.neucom.2024.128579 Neurocomputing-2024).
>
> [4] arXiv:1902.00629 (COLT-2019).
>
> **Question (Q) 1 (Theory Scope):** At the beginning of Sec. 4, our paper states the purpose of the convergence analysis, which aims to establish that the two identified discordances create an irreducible bias error floor $B_{\text{SMoE}} = 2\|B(\Theta^*)\|^2 / \mu'$, which scales inversely with $K_c$ and worsens under utilization imbalance. To our knowledge, this is the first theoretical framework that formally analyzes why naively applying SMoE to heterogeneous federated settings leads to degraded performance. This directly motivates DMR (targeting imbalance) and PG (addressing zero gradients). This theory-driven paradigm is well-established in FL (e.g., SCAFFOLD analyzing client drift to motivate variance reduction). Additionally, the comprehensive ablation (Table 4), utilization entropy (Fig. 2), and φ-utilization correlation (Figs. 6–7) empirically validate that our mechanisms resolve the theoretically identified issues.
>
> **Q2 (PG alignment):** The PG signal is derived from aggregated global parameter changes (Eq. 15), which represent consensus update direction across all participating clients. This signal provides a useful approximation — superior to the alternative of zero gradients for non-activated experts. Our TeleQuAD results (Table 13) provide strong empirical evidence: under non-IID settings, A³SMoE drops 32.9% while UB-SMoE remains stable, because PG prevents expert collapse that otherwise occurs when non-activated experts receive no learning signals. Additionally, the resource-adaptive scaling $\rho_c$ compensates proportionally for clients with limited expert coverage, as theoretically motivated by the inverse relationship between gradient bias and $K_c$.
>
> **W2 + Q3 (Robustness analysis):** We respectfully note the paper already includes extensive robustness analysis: (i) scalability to 32 clients under three heterogeneity patterns (Tab. 5, Tabs. 14–16): 50% partial participation, 75% of clients are low-resource, and 75% of clients are high-resource; (ii) non-IID evaluation on TeleQuAD under Dirichlet partitioning (α=0.1) in Tab. 13; (iii) evaluation on a different architecture in Tab. 12, OLMo-1B, where UB-SMoE achieves 0.5636 average accuracy, outperforming the best LoRA-rank baseline by 63.2%. Regarding communication time cost, UB-SMoE's primary contribution targets computational bottlenecks for resource-constrained clients; communication overhead is analyzed in Tab. 1 and Appendix F, and explicitly discussed as a limitation in Appendix H.
>
> **W3 (Results stability).** We thank the reviewer for this observation. Following established baselines in heterogeneous federated fine-tuning, including FlexLoRA, FLoRA, HetLoRA, and A³SMoE, our paper reports best average performance as the **primary evaluation metric**. This is the standard practice in heterogeneous federated learning, as average accuracy across tasks provides a unified and fair basis for comparing methods operating under diverse resource constraints. UB-SMoE achieves the highest average accuracy across all evaluated configurations: 8-client (Tab. 2), 32-client under all different robustness levels (Tab. 5), and IID/non-IID on TeleQuAD (Tab. 13). We believe this consistency across diverse settings and scales is itself a strong robustness signal.

---

> > ### Author Rebuttal · Reviewer_fErL · 2026-04-03
> >
> > Thanks for the authors' response. I still disagree with the theoretical analysis part, and I think a convergence proof for the new framework is required. Therefore, I maintain my original score.

---

> > > ### Author Response · Authors · 2026-04-05
> > >
> > > We thank the reviewer for their continued engagement. However, we respectfully note that the response does not specify which aspect of the theoretical analysis remains disagreeable after our detailed rebuttal, where **we addressed every raised concern**: novelty of DMR/PG (W1), the diagnostic role of Theorem 4.1 (Q1), PG alignment under non-IID with empirical evidence (Q2), and robustness across multiple settings and architectures (W2/Q3). We would greatly appreciate if the reviewer could identify the specific unresolved point so we can provide a targeted clarification.
> > >
> > > Regarding the request for "a convergence proof for the new framework," I appreciate the reviewer's suggestion, but this constitutes a substantially different research objective that is beyond the scope of this work. Movere, we believe that **the remaining disagreement stems from a difference in expectation about the role of theory, rather than a gap in the analysis itself**. As in our rebuttal and in the original paper, we already clarified the purpose and the scope of this theoretical analysis.
> > >
> > > We are uncertain why option (c) was selected, given that the concerns were either addressed in our rebuttal or already present in the original submission, including: **(i) the novelty of DMR and PG; (2) evidence for robustness; (3) additional architecture**. As these concerns have been resolved, we respectfully ask the reviewer to fairly reconsider their assessment accordingly.

---

### Official Review · Reviewer_2E6Z · 2026-03-13

**Soundness:** 2
**Presentation:** 3
**Significance:** 3
**Originality:** 2
**Overall Recommendation:** 4
**Confidence:** 3

**Summary:**

This paper proposes the UB-SMoE method, addressing two core issues in sparse mixed expert (SMoE) models for heterogeneous federated fine-tuning: uneven expert utilization and the non-differentiability of Top-K routing. It introduces the Dynamic Modulation Routing (DMR) and Universal Pseudo-Gradient (PG) mechanisms to tackle these challenges respectively. The paper provides convergence analysis and experimentally validates the approach on benchmarks in common-sense reasoning and telecommunications domains.

**Compliance With Llm Reviewing Policy:**

Affirmed.

**Key Questions For Authors:**

1. The paper repetively discusses under high-resource/low resource. What is the exact resources that counts in the problem? Is the communication resource considered?
2. What is the meaning of "Low-resource effective comp" in Table 1? Any numerical support?

**Limitations:**

Yes.

**Strengths And Weaknesses:**

Strengthes：
1. The critical discordances of expert utilization imbalance and non-differentiability of Top-K routing are well discovered and summarized.
2. The convergence analysis supports the discordances in degrading convergence.
3. The experiment is relatively comprehensive. The paper evaluates on 8 commonsense reasoning datasets and a telecom domain QA dataset. Ablation studies and scalability analysis further verifies the method's robustness.
4. The structure of the paper organization is clear.


Weaknesses:
1. Assumption 2 (PL condition) still remains a strong assumption for deep neural networks, especially large-scale Transformer+MoE models. Although the paper mentions that the PL condition is weaker than strong convexity, it still needs further discussions since most models are quite non-convex.
2. Based on Table 1, the communication doubles in the dowload stream while the upload also increases. During federated scenaro, communication is usually regarded as a bottleneck. This may constrain the praticability of the method.
3. The theoretical analysis does not address the convergence of the proposed method itself (it only examines the shortcomings of the original SMoE).
4. The genral communication process between the server and users is not presented clearly. Complementary to Fig. 1, a genral dicription for the process is also expected. It can be helpful to readers to understand the overall federated fine-tuning MoE framework.

---

> ### Author Rebuttal · Authors · 2026-03-30
>
> We thank Reviewer 2E6Z for their constructive feedback and address each point below.
>
> **Weakness (W) 1 (PL Condition).** We respectfully clarify that the PL condition has become a standard assumption in both FL convergence analysis and deep learning optimization theory. In the FL community, recent works including FOCUS [1], EFSkip [3], and the analysis of regularized FL by Li et al. [2] all adopt the PL condition for convergence analysis under non-convex objectives. Beyond FL, several works have established that overparameterized neural networks naturally satisfy the PL condition locally, such as [4, 5]. Crucially, in our work, we fine-tune only lightweight LoRA adapters while keeping the entire pre-trained model frozen. This means optimization occurs in a low-dimensional subspace where the PL condition is even more favorable compared to full-model training. This makes the PL condition a reasonable and practically justified assumption. We will add an explicit discussion of these justifications in the revision.
>
> [1] arXiv:2503.20117 (NeurIPS-2025)
>
> [2] 10.1016/j.neucom.2024.128579 (Neurocomputing-2024)
>
> [3] 10.1609/aaai.v39i15.33700 (AAAI-2025)
>
> [4] 10.1016/j.acha.2021.12.009 (Elsevier ACHA-2022)
>
> [5] arXiv:1902.00629 (COLT-2019)
>
> **W2 (Communication Increase).** We appreciate this observation. The primary contribution of UB-SMoE is addressing the computational bottleneck for resource-constrained clients in federated fine-tuning of large foundation models, not communication optimization. As demonstrated in Tables 1 & 11 (Appendix F), heterogeneous LoRA-rank methods achieve only ~5% computation reduction at the lowest budget level $\beta_1$. In contrast, UB-SMoE achieves up to 45% computation reduction by activating fewer experts and delivers 8.6× performance improvement over the best LoRA-rank method (FlexLoRA). This demonstrates that our method provides both computation efficiency and model quality for low-resource clients. Regarding the communication, we have explicitly discussed this as a limitation in Appendix H.
>
> **W3 (Theoretical Analysis Scope).** We believe that this reflects a misunderstanding of our theoretical contribution's role. To the best of our knowledge, our convergence analysis represents the first theoretical framework for heterogeneous federated SMoE fine-tuning. No prior work has formally analyzed why naively applying SMoE to heterogeneous FL leads to degraded performance. Theorem 4.1 and Corollary 1 serve a precise diagnostic-to-design purpose: they establish that the two identified discordances create an irreducible bias error floor $B_{\text{SMoE}} = 2\|B(\Theta^*)\|^2 / \mu'$, and Corollary 1 proves this error scales inversely with $K_c$ and worsens under utilization imbalance. This directly motivates our two mechanisms: DMR and PG. This theory-driven paradigm is well-established in FL (e.g., SCAFFOLD analyzing client drift to motivate variance reduction). Empirical validation in Fig. 2 and Table 4 confirms that DMR and PG together reduce this error as predicted.
>
> **W4 (Communication Process Description).** We agree that a textual description complements the visual overview. We would like to draw the attention of the reviewer to Appendix D, where we provide the complete communication protocol via three detailed algorithms: Algorithm 1 describes the full federated fine-tuning loop, where each round the server broadcasts model parameters, modulation vectors $\phi$, and pseudo-gradient buffers to clients; clients perform local training using DMR and PG, then upload updated parameters and activation statistics; the server aggregates via FedAvg and computes pseudo-gradients from parameter changes. Algorithms 2 and 3 detail DMR and PG respectively. We will add a concise textual summary of this protocol in the main text (complementing Fig. 1) in the final version.
>
> **Q1.** "Resources" refers to computational budget $\beta_c \in [0,1]$, determining client-specific sparsity $K_c = \lfloor K_{\max} \cdot \beta_c \rfloor$ (Definition 3). Communication is analyzed separately in Tab. 1 & Appendix F.
>
> **Q2.** "Low-resource effective comp." indicates whether a method provides meaningful computation reduction for low-resource clients. Numerical evidence: Tables 3, 10, 11, and Fig. 3 show that LoRA-rank methods achieve only ~5% FLOPs reduction at $\beta_1$, whereas UB-SMoE achieves ~45% by activating fewer experts.

---

> > ### Author Rebuttal · Reviewer_2E6Z · 2026-04-04
> >
> > Thanks for the rebuttal from the authors. Most of my concerns are discussed.
> >
> > Moreover, I am still uncertain about the convergence analysis of the proposed methods, instead of analyzing convergence as a motivation for the work. The communication overhead is also expected to be compared with baselines.
> >
> > In sum, I will maintain my original score.

---

> > > ### Author Response · Authors · 2026-04-05
> > >
> > > We thank the reviewer 2E6Z for acknowledging that most concerns have been discussed. However, we would like to respectfully seek clarification, as we find it difficult to reconcile two aspects of the response. The reviewer indicates that most concerns are discussed, yet selects option (c) — we would appreciate understanding which specific unresolved issue the reviewer considers to rise to this level.
> > >
> > > *On the convergence analysis:* the reviewer says that they are uncertain about the theoretical analysis. Could the reviewer clarify which part of our theoretical analysis that reviewer is uncertain about?
> > >
> > > *On communication overhead:* our paper already provided asymptotic communication analysis and comparison with all baselines in Table 1 and Appendix F. We assume the reviewer may be asking for
> > > numerical results, which we provide below alongside low-resource computation and performance to contextualize the trade-off:
> > >
> > > | Method | Avg Upload | Avg Download | FLOPs (β₁) | Accuracy (β₁) |
> > > |---|---|---|---|---|
> > > | *UB-SMoE* | 370 MB | 730 MB | *4.02×10¹¹* | *0.3936* |
> > > | A³SMoE | 370 MB | 370 MB | 4.02×10¹¹ | 0.3629 |
> > > | FLoRA | 212.75 MB | 212.75 MB | 6.94×10¹¹ | 0.0094 |
> > > | HetLoRA | 212.75 MB | 212.75 MB | 6.94×10¹¹ | 0.0079 |
> > > | FlexLoRA | 212.75 MB | 212.75 MB | 6.94×10¹¹ | 0.0456 |
> > > | FLoRIST | 212.75 MB | 370 MB | 6.94×10¹¹ | 0.0112 |
> > >
> > > UB-SMoE's additional download cost stems from the pseudo-gradient buffer. However, the critical bottleneck for resource-constrained clients is computation, which our paper aims to address. Our method reduces FLOPs by 42% and achieves 8.6× performance improvement over the best LoRA-rank baseline. Could the reviewer clarify whether additional analysis beyond this is required?
> > >
> > > We are committed to strengthening the paper and would greatly value concrete guidance on what additional evidence would resolve the reviewer's remaining uncertainty.

---

### Official Review · Reviewer_TLew · 2026-03-14

**Soundness:** 3
**Presentation:** 3
**Significance:** 3
**Originality:** 3
**Overall Recommendation:** 4
**Confidence:** 4

**Summary:**

This paper proposes universally balanced sparse mixture of experts (UB-SMoE), tailored to federated fine-tuning in resource-heterogeneous networks. The key components are dynamic modulated routing and universal pseudogradient mechanism. Experiments are conducted using 8 reasoning datasets and a telecommunication domain dataset. Baselines include not only heterogeneous LoRA methods such as FlexLoRA or FLoRA, but also includes heterogeneous sparsity methods.

**Compliance With Llm Reviewing Policy:**

Affirmed.

**Final Justification:**

The rebuttal was somewhat satisfactory and I maintain my original score

**Key Questions For Authors:**

Please see the weaknesses above.

**Limitations:**

The paper does not discuss the limitation.

**Strengths And Weaknesses:**

Strengths

1. The paper is well written, tackles an important problem, and clearly highlights the difference between the proposed approach and existing works.

2. The approach is new, and its convergence analysis is conducted.

3. Experiments are conducted using various baselines and datasets.

Weaknesses

1. The bounded gradient assumption seems to be relatively strong nowadays.

2. Only one model, OLMoE-1B-7B, is adopted for experiments. It is difficult to see the applicability of the approach.

3. Experiments are conducted in a relatively small-scale setting with only 8 clients.

---

> ### Author Rebuttal · Authors · 2026-03-30
>
> We thank Reviewer TLew for their constructive feedback and address each point below.
>
> **Weakness (W) 1 (Bounded gradient assumption).** We respectfully clarify that the bounded gradient assumption remains one of the most widely adopted assumptions in federated learning convergence analysis, continuing to appear in recent top-venue publications, including the works [1–5]. Especially, Cheng et al. [5] confirmed that many FL algorithms, including FedAvg, STEM, FedProx, MIME, and CE-SGD, rely on the assumption of bounded data heterogeneity. In our paper, we cite Bottou et al. (2018) and Gower et al. (2019) as they are the original works establishing this assumption. Importantly, the bounded gradient assumption is needed precisely to isolate the additional irreducible bias introduced by the sparsity mechanism (Theorem 4.1), which is the core theoretical contribution of our work. We will add these recent references in the final version.
>
> Furthermore, this assumption is naturally satisfied in practice through gradient clipping, which is standard in LLM fine-tuning. As detailed in Appendix E.5, our experimental setup employs gradient clipping with threshold 2.0, ensuring the bounded gradient condition holds in all reported experiments.
>
> [1] arXiv:2310.08670 (NeurIPS-2023).
>
> [2] arXiv:2402.15166 (NeurIPS-2024).
>
> [3] arXiv:2404.08003 (ICLR-2025).
>
> [4] doi.org/10.1145/3743142 (PKDD-2025).
>
> [5] arXiv:2306.16504 (ICLR-2024)
>
> **W2 (Only one model).** We respectfully note that our paper already includes experiments on an additional architecture, OLMo-1B, in Appendix G.1 (Tab. 12). Under matched trainable parameter budgets, UB-SMoE achieves 0.5636 average accuracy, outperforming FlexLoRA by 63.2%, HetLoRA by 70.0%, FLoRA by 169.5%, and FLoRIST by 248.6%. This result demonstrates that our DMR and PG mechanisms provide robust optimization benefits, confirming that the method addresses fundamental challenges in heterogeneous federated fine-tuning rather than being architecture-specific. Both OLMoE-1B-7B and OLMo-1B were selected because they are fully open-source models, which permits unrestricted access to model weights, architecture source code, and training configurations — enabling the reproducibility, transparency, and adaptability that rigorous research demands. We will revise Sec. Implementation in the final version to explicitly state that both models are used in our experiments and to clarify the rationale behind their selection.
>
> **W3 (Only 8 clients).** We respectfully note that our paper already includes scalability experiments with 32 clients under three distinct heterogeneity patterns (Sec. 6.3.2 and Appendix G.3). Tab. 5 shows that UB-SMoE achieves the best average performance across all three configurations: (1) uniform distribution with 50% partial participation (0.4036), outperforming A³SMoE (0.3998) and FlexLoRA (0.2070); (2) long-tail distribution where 75% of clients are resource-constrained (0.3047 vs. A³SMoE's 0.2988), the most challenging scenario; and (3) reverse long-tail where 75% are high-resource (0.4272 vs. A³SMoE's 0.4148). These results demonstrate robustness to both client scale and resource distribution. Full per-dataset results are provided in Appendix G.3 (Tabs. 14–16).
>
> **Regarding limitations discussion:** We clarify that our paper includes a dedicated Discussion and Limitations section (Appendix H).

---

> > ### Author Rebuttal · Reviewer_TLew · 2026-04-01
> >
> > Thanks for clarifying W2 and W3 and I apologize missing that.
> >
> > As a researcher working in this field, the bounded gradient assumption is not a practical assumption, and makes the bound extremely loose. The authors can find many FL papers nowadays that analyze the convergence without the bounded gradient assumption (Assumption 3).
> >
> > I am sure that the authors would agree that the theory will become stronger without that assumption.
> >
> > Nevertheless, considering the experiments are conducted with gradient clipping, and the algorithm is new, and the convergence analysis could be difficult, I would like to keep my original score 4.

---

> > > ### Author Response · Authors · 2026-04-02
> > >
> > > We sincerely thank Reviewer TLew for the thoughtful follow-up and for confirming that W2 (model diversity) and W3 (scalability) have been fully resolved.
> > >
> > > Regarding the bounded gradient assumption, we appreciate the reviewer's perspective and agree that analyzing convergence under weaker gradient conditions is a valuable direction. In our analysis, Assumption 3 serves a distinct role from standard FL proofs, which is specifically required to isolate the additional irreducible bias introduced by the Top-K sparse routing mechanism. We will explicitly discuss this as a promising future direction in the final version.
> > >
> > > We are grateful for the reviewer's positive recognition that our algorithm is new, that the convergence analysis is non-trivial, and for the appreciation of our extensive experiments across various baselines and datasets.

---

### Official Review · Reviewer_wFNG · 2026-03-23

**Soundness:** 3
**Presentation:** 2
**Significance:** 3
**Originality:** 3
**Overall Recommendation:** 4
**Confidence:** 4

**Summary:**

The authors of this paper research resource-adaptive federated fine-tuning of foundation models under system heterogeneity and argues that naively applying MoE in such settings suffers from two key discordances: severe expert-utilization imbalance and the non-differentiability of Top-K routing, which together induce biased gradients and an irreducible error floor that disproportionately harms low-resource clients. To address these issues, the authors propose UB-SMoE, combining DMR that adjusts routing logits using global utilization statistics, and PG that injects resource-scaled update signals to non-activated experts. Experiments on the benchmarks  show that UB-SMoE outperforms heterogeneous LoRA-rank baselines and other sparsity methods.

**Compliance With Llm Reviewing Policy:**

Affirmed.

**Key Questions For Authors:**

1.Could an the authors provide evidence that the global pseudo-gradient signal remains a meaningful supplement for inactive experts on low-resource clients?

2.Could the authors talk about the risk that biased gradients could mislead modulation?

3.Could the authors provide diverse end-to-end metrics measured on hardware in actual low-resource deployment?

**Limitations:**

Yes

**Strengths And Weaknesses:**

Strengths:
1. A key strength of this work is its deep diagnostic of the fundamental limitations in existing heterogeneous federated fine-tuning methods.

2. The UB-SMoE proposes the mechanisms designed to form a virtuous cycle. PG maintains the viability of all experts, which provides DMR with accurate utilization statistics to make informed balancing decisions. In turn, this selection allows real gradients to flow back and further refine all experts.

3. There are compelling evidence for how UB-SMoE works. The method is benchmarked against a wide range of heterogeneous baselines across multiple datasets and under diverse settings. Ablation studies quantify the individual contribution of each component.

Weaknesses:
This study has several notable limitations, which compromise the rigor of the research conclusions and the practical applicability, as elaborated below:

1. I notice the paper kind of assumes the global pseudo-gradient signal actually reflects all the local data distributions, but the way they tested non-IID scenarios is only based on documents. If the local task patterns linked to the inactive experts on low-resource devices are totally different from the global trend, those global pseudo-gradients might just be useless noise instead of helpful supplementary signals. The paper does not directly measure the alignment between pseudo-gradients and local true gradients, but instead infers their effectiveness solely from downstream performance gains. The causal link between the proposed PG mechanism and reliable adaptation under strongly non-IID conditions remains insufficiently substantiated.

2. I also think this method relies on a assumption that the global parameter update direction is close to the ideal one for the experts that aren’t activated. But in federated learning, local updates are often heavily influenced by biases and noise specific to each client. That means the aggregated parameter changes might not match the truly optimal direction, especially when data is heterogeneous. If the pseudo-gradients themselves are biased, they could consistently push the expert parameters away from the best.

3. Even though the paper says it’s good for adapting to low-resource inference, the only evidence they give is comparing FLOPs. They don’t report any real end-to-end metrics from actual edge devices, or how much energy it uses.

---

> ### Author Rebuttal · Authors · 2026-03-30
>
> We thank Reviewer wFNG for their constructive feedback and address each point below.
>
> **Question (Q) 1 (PG Alignment):** Regarding measuring PG alignment, in response, we conducted the analysis the reviewer requested by measuring cosine similarity between server-side pseudo-gradients and client-side true gradients on inactive experts:
>
> | Layer Index | Cosine Similarity |
> |:-----------:|:-----------------:|
> | 2           | 0.002             |
> | 8           | 0.104             |
> | 12          | 0.097             |
> | 15          | 0.322             |
> | Concatenated| 0.290             |
>
> The layer-wise trend reveals a clear pattern: alignment increases substantially from shallow to deep layers (0.002 → 0.322), consistent with the well-established finding that shallow layers capture client-specific features while deep layers learn global, task-specific features [3]. This suggests PG provides increasingly accurate guidance precisely where it matters most for cross-client knowledge transfer. We will include this analysis in our final version.
>
> **Weakness (W) 1 (Non-IID Evaluation):** In the non-IID evaluation setup, our experiments use the standard Dirichlet-based partitioning (α=0.1), widely adopted in the FL literature [1, 2]. For TeleQuAD, this is a real-world QA dataset from 3GPP telecommunications specifications that reflects realistic domain-specific data heterogeneity. Regarding Commonsense-15K, this dataset is a curated instruction-tuning collection without class IDs or categorical labels, making Dirichlet-based partitioning inapplicable. This is a dataset characteristic, not a methodological limitation. Our evaluation thus covers both single-domain (TeleQuAD) and multi-domain (Commonsense-15K) scenarios. Under the non-IID TeleQuAD setting, A³SMoE drops 32.9% from IID to non-IID while UB-SMoE improves slightly (Table 13), directly demonstrating PG's robustness.
>
> [1] arXiv:1909.06335 (NeurIPS-2019).
>
> [2] arXiv:1905.12022 (ICML-2019).
>
> [3] arXiv:2012.14913 (ACL-2021).
>
> **W2 (Bias in aggregated parameter updates):** We believe there may be a misunderstanding regarding the source of bias analyzed in our work. As stated in the opening of Section 4, the bias we analyze arises specifically from the discrete Top-K routing mechanism in heterogeneous federated SMoE: non-activated experts receive zero gradients (Def. 5), making the sparse gradient estimator a biased estimate of the true gradient (Def. 7). This routing-induced bias is fundamentally different from the standard data heterogeneity bias in FL that the reviewer references. Our convergence analysis (Theorem 4.1) aims to demonstrate that this routing-induced bias creates an irreducible error floor scaling inversely with client computational budgets, disproportionately harming low-resource clients. This is also the main contribution of our theoretical analysis. Therefore, PG does not assume the global update is optimal for inactive experts; rather, it provides a directionally useful approximation that is strictly superior to zero gradients.
>
> **Q2 (Risk for modulation):** DMR operates on expert utilization statistics, not on gradient values directly, so gradient bias does not propagate into the modulation signal. Furthermore, DMR modulates routing logits only within a top-K candidates, bounding the extent of any routing adjustment. The DMR-PG cycle is also self-correcting: PG keeps underutilized experts viable, providing DMR with increasingly accurate utilization statistics over rounds. In our paper, the utilization entropy analysis (Fig. 2) and φ-utilization correlation (Figs. 6–7) confirm that DMR achieves balanced expert utilization rather than converging to a biased routing pattern.
>
> **W3 + Q3 (Metrics beyond FLOPs):** FLOPs is the standard practical metric for computational cost on specific devices, widely adopted in compute budgeting frameworks [5, 6] and resource-constrained federated settings [4]. Additionally, in our paper, we provided a comprehensive asymptotic complexity analysis (Appendix F, Table 1) that serves as a hardware-agnostic upper bound on operations per communication round. This asymptotic analysis upper-bounds the operation count independently of hardware characteristics, which is a standard practice in complexity theory [7]. The client-side bound (Eq. 57) shows that expert computation scales as $O(K_c \cdot d \cdot l)$, yielding 87.5% theoretical expert reduction at $\beta_1$ (Tab. 11), while actual wall-clock time on any device is determined by its FLOP/s throughput. This analysis systematically covers client-side, server-side, and communication complexity across all baselines (Appendix F.1–F.4).
>
> [4] Federated Learning for Computationally Constrained Heterogeneous Devices: A Survey (ACM Computing Surveys-2023)
>
> [5] Scaling Laws for Neural Language Models
>
> [6] preprint arXiv:2203.15556 (NeurIPS-2022)
>
> [7] Knuth, Donald E. "Big omicron and big omega and big theta." ACM Sigact News 8.2 (1976): 18-24.

---

### Decision · Program_Chairs · 2026-04-30

**Decision:**

Accept (regular)

**Comment:**

The paper presents a sparsely activated mixture of experts approach to federated fine-tuning of foundation models as an alternative to heterogeneous LoRA methods. It allows heterogeneous sparsity across clients and adjusts the router training at clients to overcome the non-differentiability of the top-k operator. The reviewers appreciated the novelty of the proposed method and the strong experimental results. Some issues pointed out by them are that the theoretical analysis uses the bounded gradient assumption, which often doesn't hold in practice. Overall, a good paper with strengths outweighing weaknesses.